# Simple noise estimates and pseudoproxies for the last 21k years

Oliver Bothe[1], Sebastian Wagner[1], and Eduardo Zorita[1]

[1]Helmholtz Zentrum Geesthacht, Institute of Coastal Research, 21502 Geesthacht, Germany

**Correspondence:** Oliver Bothe (ol.bothe@gmail.com)

**Abstract.** Climate reconstructions are means to extract the signal from uncertain paleo-observations, so called proxies. It is essential to evaluate these reconstructions to understand and quantify their uncertainties. Similarly, comparing climate simulations and proxies requires approaches to bridge the temporal and spatial differences between both and address their specific uncertainties. One way to achieve these two goals are so called pseudoproxies. These are surrogate proxy records within the virtual reality of a climate simulation. They in turn depend on an understanding of the uncertainties of the real proxies including the noise-characteristics disturbing the original environmental signal. Common pseudoproxy approaches so far concentrate on data with high temporal resolution over the last approximately 2,000 years. Here we provide a simple but flexible noise model for potentially low-resolution sedimentary climate proxies for temperature on millennial time-scales, the code for calculating a set of pseudoproxies from a simulation, and one example of pseudoproxies. The noise model considers the influence of other environmental variables, a dependence on the climate state, a bias due to changing seasonality, modifications of the archive (for example, bioturbation), potential sampling variability, and a measurement error. Model, code, and data allow to develop new ways of comparing simulation data with proxies on long time-scales. Code and data are available at https://doi.org/10.17605/OSF.IO/ZBEHX.

## 1 Introduction

Proxy-records and derived reconstructions are our only observationally based information about past climates before the period covered by human observations, that is before we have documentary or instrumental evidence. Climate reconstruction methods statistically process the information in the proxy records to extract the recorded climate signal. However, this climate signal is potentially multivariate, and we are often only interested or able to extract the signal for one single climatic parameter.

All other imprints of climate are noise relative to this variable of interest. Furthermore, part of the variability in the proxy records is not caused by the climate but other factors influencing the original generation of the proxy-record. Thus, there are climatic and non-climatic noise contributions to the proxy variability. This proxy noise may cause biases and uncertainties in the resulting climate reconstructions. Evaluating the quality and reliability of reconstructions and of proxy-records requires an understanding of the noise in the proxy-records. Only this knowledge allows us to obtain reliable estimates of the errors in reconstructed properties.

Some aspects of statistical climate reconstruction methods can be evaluated in so-called pseudoproxy experiments. In these experiments, the reconstruction methods are mimicked for example in the controlled conditions provided by climate simula-

tions with Earth System Models. However, for these tests surrogate proxy records have to be produced, which are compatible with the climate simulated by these models—the pseudoproxies. In testing the reconstruction methods, pseudoproxies eventually replace the real paleo-observations in the method and the virtual climate of the Earth System simulation stands in for the real climate.

Our use of the term pseudoproxy follows the literature since Mann and Rutherford (2002). That is, a pseudoproxy represents a modification of observational data, reanalysis data, or simulation output. It replaces real world proxies in an application. The term does not necessarily refer to substitutes for specific proxy records or particular proxy types. That is, the term pseudoproxy does not by itself imply that the modifications of the input data represent validly the uncertainties or characteristics of real world data. This view of the term pseudoproxy is in line with the past literature (compare, for example, Mann and Rutherford,

2002; Osborn and Briffa, 2004; Von Storch et al., 2004; Jones et al., 2009; Graham and Wahl, 2011; Thompson et al., 2011; Lehner et al., 2012; Smerdon, 2012; Hind et al., 2012; Annan and Hargreaves, 2013; Kurahashi-Nakamura et al., 2014; Steiger and Hakim, 2016). Modifications of the input data may be as simple as adding white or colored noise or they may invoke more complex forward approaches (for example mechanistic Proxy System Models, Evans et al., 2013, see below).

Studies of the climate of the past 2,000 years regularly use such pseudoproxy approaches mimicking annually resolved

proxies such as dendroclimatogical ones. Smerdon (2012) reviews the approach of using pseudoproxy-experiments to evaluate reconstruction methods with a focus on the last millennium. Such methods basically originated in the work of Mann and Rutherford (2002) focussing on climate-field reconstructions. The review by Smerdon (2012) emphasizes the essential contribution of pseudoproxy-experiments to our understanding of past climates and to evaluating our methods of studying past climates. To date, most studies using pseudoproxies concentrated on the last few millennia. Few studies considered periods

further in the past (e.g., Laepple and Huybers, 2013; Dolman and Laepple, 2018; Dee et al., 2018).

For a useful test of reconstruction methods, the pseudoproxies should be as realistic as possible, with statistical properties similar to the real proxies. This is achieved by contaminating the climate variables simulated by the Earth System Model with statistical noise with a certain amplitude and statistical characteristics. These properties ideally are based on estimates of a realistic or at least plausible noise to successfully mimic the behavior of real-world proxies.

In our understanding there are various approaches to obtain such pseudoproxies. These range from most comprehensive to most simplified. We can try to obtain a comprehensive representation of a so called proxy system (Evans et al., 2013) from the environmental influences on a sensor to our measurement and formulate this into a mechanistic forward model of the system of interest. Such models can be very complex or they may concentrate solely on a core set of processes (compare the full and reduced implementations of the Vaganov-Shashkin approach to modelling tree-rings presented by Evans et al., 2006;

Tolwinski-Ward et al., 2011). That is, the first approach to obtaining pseudoproxies is process-based. Other, more reduced approaches potentially ignore this mechanistic process understanding and focus on stochastic expressions of the noise that influence our inferences about past climates. Such an approach can try to formulate mathematically tractable expressions for statistical noise-terms, which represent the different processes or effects influencing the stages from the original environmental conditions to our final observation [Dolman et al., in preparation, A. Dolman, personal communication, 2018, T. Laepple,

personal communication, 2017]. Another way of producing pseudoproxies by focussing on stochastic noise expressions uses

simple estimates of plausible errors. These different approaches can be specific for certain proxy types or very general. They can focus on one stage of the proxy system from environment to measurement or consider multiple stages.

All these approaches fit into the concept of a proxy system model as described by Evans et al. (2013). The idea of forward models to study the behavior of proxies and proxy systems is not new (e.g., Schmidt, 1999; Tolwinski-Ward et al., 2011; Thompson et al., 2011) but Evans et al. (2013) were the first to clearly delineate the modelling of proxy systems. A proxy system represents the biological, chemical, geological, and possibly also documentary system that translates environmental influences into an archived state on which researchers make observations. We usually refer to these observations when speaking of climate proxies. A proxy system model is a representation of how the proxy system translates the environmental influences into our observations based on our understanding. Evans et al. (2013) present a generalized concept of this modelling approach, which consists of three components: First, a sensor model reacts to the environmental influences. Second, an archive model transforms these sensor recordings into archive units. A third model translates the archive into representations of what we usually observe on an archive. For example, the sensor 'tree' records the environmental influences in its archive 'wood', and we can make measurements on this archive in form of tree-ring counts and widths etc. The full system from recording to observation is the proxy system.

Each stage in this system and its model representations adds uncertainty, and each stage omitted in a generalization also increases uncertainty. For example, the environment and the final reconstruction process can be additional stages, but we can try to include the associated uncertainties in any of the three stages proposed by Evans et al. (2013). That is, considering the reconstruction stage, the calibration introduces additional uncertainty, which is not a priori captured by the stages sensor, archive, and measurement. We can argue to include this additional source of error in the measurement stage. We can also argue that these uncertainties are de facto uncertainties resulting from processes at the sensor stage or at the archiving stage and include them there. Similarly, the sensor model does not necessarily account for all uncertainties of the environmental influences. An additional environmental stage could provide weighted data of various environmental influences (compare, e.g., Dee et al., 2018). These processes, however, can also be included in the sensor model or uncertainties can be assumed to mostly affect the measurement model. In short, inferences about past climates from proxy-data are based on observations on an archive that accumulated a property of a system. This (property of the) system was sensitive to and recorded an environmental process at some date. From the recording stage to our inference there are multiple sources of error to our inference.

The potential errors include different sources of noise related to laboratory uncertainties like measurement errors and reproducibility, local disturbances, dating uncertainty, time resolution, serial autocorrelation, and all possibly dependent on the overall climate state. Further uncertainty includes habitat preferences, seasonal biases, the variability in the relation between sensor and environment, long term changes in this relation, long term modifications of the archive, sampling variability and sampling disturbances, and not least generally erroneous assumptions on the researcher's side on the relation between recording sensor and environment, i.e., the calibration relation. A recipe for calculating pseudoproxies may include potential error estimates not only for parts of the assumed proxy-system but also for the relation between the 'observed' data and time, that is the anchoring of the data in time.

Regarding dating/age uncertainty, there are various approaches to dealing with it (e.g., Breitenbach et al., 2012; Carré et al., 2012; Anchukaitis and Tierney, 2013; Comboul et al., 2014; Goswami et al., 2014; Brierley and Rehfeld, 2014; Rehfeld and Kurths, 2014; Kopp et al., 2016; Boers et al., 2017) of which a number try to transfer the dating uncertainty towards the proxy-record-uncertainty (e.g., Breitenbach et al., 2012; Goswami et al., 2014; Boers et al., 2017). Our interest explicitly is to include the uncertainty from the dating in an statistical noise term for a pseudoproxy time-series. Therefore, we do not consider Bayesian or Monte Carlo methods but take a simple approach to develop an error term for the uncertainty in the dating. We also do not include explicit age-modelling (compare, e.g., Haslett and Parnell, 2008; Blaauw and Christen, 2011; Trachsel and Telford, 2017).

Besides evaluating reconstruction methods, a plausible estimate of noise within the proxies also can assist in comparison studies between model-simulations and the proxy-records or among different model-simulations. This increases our understanding about past climate changes by consolidating information from all available sources, which are proxy records and model simulations. The lack of high-quality observations with small uncertainty is always going to hamper efforts to assess the quality of model-simulations of past climates. Such comparisons have to rely on the paleo-observations from proxies, and even the highest-quality proxy-records have an irreducible amount of uncertainty. Most often data-model-comparisons take place in the virtual reality of the model and use the modelled variables. In the case of proxies, the comparison is between, for example, a temperature reconstruction and a model. The alternative is to compare both in the proxy-space using a proxy-representation of the model-climate. Pseudoproxies or a recipe how to compute them may be part of an interface between the data on the one side and the model simulations on the other side.

Recent years saw an intensification in the research on forward modelling proxies for understanding proxies, testing reconstruction methods, and evaluating simulation output (see, for example, Dolman and Laepple, 2018; Dee et al., 2015, 2018; Konecky et al., 2019). Many of these approaches follow the concept of considering sensor, archive, and observations as distinct steps in the process. Still, few of these approaches consider transient time-scales beyond the late Holocene. Nevertheless, particularly the work by Dolman and Laepple (2018) and also Dee et al. (2018) allow for the calculation of different sedimentary proxies over multi-millennial time-scales based on knowledge of certain processes in the respective proxy systems.

In this paper, we adopt the conceptual sub-divisions of Evans et al. (2013) to present a formal but still simple noise based approach to describe the disturbances masking the signal in proxy records. This approach can also be applied to produce pseudoproxies for timescales longer than the last few millennia, that is proxies with coarser time resolutions than interannual and afflicted by larger degrees of dating uncertainty. Thereby this work extends on previous pseudoproxy-approaches, which often concentrated on well dated proxy-systems affected by fewer sources of uncertainty.

The following presents a set of assumptions on proxy noise and estimates for some of the mentioned error sources. We further provide pseudoproxies based on these assumptions for the TraCE-21ka simulation (Liu et al., 2009), which covers the last 21,000 years. We concentrate on proxies, which are subject to some kind of sedimentary process. Thus, our work appears to be particularly similar to the proxy system model for sedimentary proxies implemented by Dolman and Laepple (2018). Dolman and Laepple (2018) also consider the long time-scales since the last glacial maximum and rely on output from the TraCE-21ka simulation for their forward modelling. Both, the present manuscript and Dolman and Laepple (2018) follow the

concept outlined by Evans et al. (2013). The main difference between Dolman and Laepple (2018) and the present study is that they provide a simple process-focussed model of the proxy system, whereas we try to provide a simple characterisation of the noise in the proxy system that finally influences the proxies. The process-based formulation of Dolman and Laepple (2018) concentrates on two types of marine proxies whereas our noise-based approach tries to generalize over sedimentary proxy types. We regard both approaches as complementary and want to emphasize the value in having a multitude of methods to assess model-simulations and reconstruction methods.

Our approach contributes to the existing proxy system modelling and pseudoproxy computation applications by being an intermediate step between complex forward modelling approaches and the noise based approaches, of which the latter may ignore the proxy system workings. Our code simplifies and generalizes more complex assumptions. The noise-focus and the generalizations allow us to provide global pseudoproxy data and an ensemble of pseudoproxy data using the TraCE-21ka simulation over the time-scale of the last 21 thousand years. The manuscript assets at https://doi.org/10.17605/OSF.IO/ZBEHX provide the generated pseudoproxy data and also include sample code. Thereby the manuscript provides for one simulation the data to make an informed comparison with real proxies and the data to evaluate reconstruction techniques. Code and assumptions enable any interested user to produce similar pseudoproxies for their simulation of interest. We consider the measurement error, local changes to the original proxy-recording (compare, e.g., Laepple and Huybers, 2013), the basic climate state, a potential bias, and a simple estimate of the effect of dating uncertainty. All noise expressions are coded in a way to flexibly allow for different colors and types of noise.

## 2 Input Data

We use the annual mean temperature at each grid-point of the TraCE-21ka simulation (Liu et al., 2009). To date, this is the only available interannual transient Earth System Model simulation covering the last 21,000 years. Specific technical considerations, for example, related to freshwater pulses and sea-level adjustments lead to some artefacts in the simulation output data fields. A brief description of the simulation can be found at http://www.cgd.ucar.edu/ccr/TraCE/, and the Ph.D.-dissertation of He (2011) provides more details.

The presented results and Figures are generally for one grid-point at 150°E, 38.97°N. The simulation output at this grid-point has the benefit of representing a rather smooth evolution of temperature over the last 21,000 years. On the other hand, the less extreme climate variations to be captured in a subsequent pseudoproxy can be seen as a disadvantage. The document assets provide Figures equivalent to those in this document for a grid-point at 105°W, 45.39°S in the South Pacific.

On multi-millennial time-scales we have to consider changes in the insolation caused by changes in earth's orbital elements. Global insolation data is calculated using the R (R Core Team, 2017) package palinsol (Crucifix, 2016). We use for most noise-processes simple Gaussian noise. However, as the code is flexible, the user can easily change this.

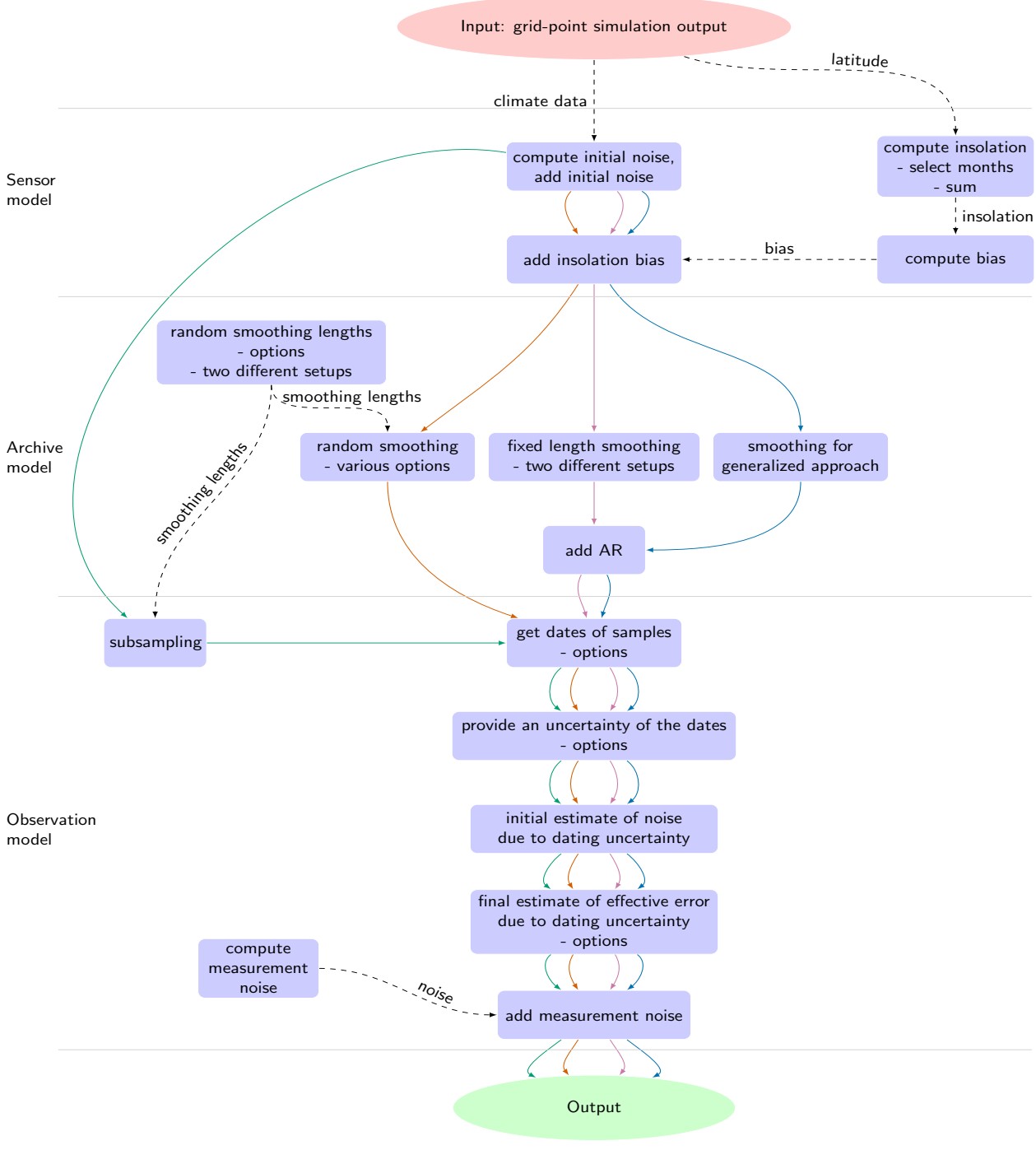

**Figure 1.** Conceptual flow of the procedures.

## 3 Considerations and Results

In defining what we consider as noise, we first have to state the signal, which we assume the proxy system records. That is, do we assume that the proxy records local or regionally accumulated signals? Here, we take the signal of interest to be local, that is non-local influences enter the noise term and are not part of the signal. In addition, there are further local factors which affect the recording of the signal but are not part of the signal of interest. The appendix provides tables (Tables A1 to A4) summarising the considered parameters and noise models in the various steps of our considerations.

In the following, we distinguish between different sources of errors related to the concepts of sensor, archive, and measurements of Evans et al. (2013). Figure 1 summarises our procedure. Each section contains a discussion of the implications of the respective error term. Afterwards we discuss the results of applying the respective step in the framework to the output of the TraCE-21ka simulation.

### 3.1 Assumptions on essential error sources 1: Sensor

#### 3.1.1 Noise and bias

The sensor, that is for example an organism or a physical or biogeochemical process, reacts to multiple parts of its environment. Researchers' interest often is only in one of the environmental variables. The sensor, $S$, is likely a nonlinear function of the environment, $S(E)$, where $E = \{e_i\}$, with $e_i$ being components of the environment. If our interest is only in the sensor's reaction to one variable, $T$,

$$S(E) \approx \widehat{S}(T, \eta_i) \tag{1}$$

Under this assumption, further components of the environment besides $T$ contribute only noise components $\eta_i$ to the reaction of the sensor. Errors due to noise are not necessarily additive but can also be multiplicative or could bias the estimate. In a first step we, here, assume the sensor-reaction to be

$$S(E) \approx \widehat{S}(T) + \eta_i \tag{2}$$

Any of these errors or noise-processes may show auto-correlation in either space or time or both. Any such process may, in turn, add memory to the sensor-system. Indeed this memory-effect and spatial or temporal correlations may be large. For example, if a process takes place in an environment with slowly and fast varying components, and our interest is in one of the fast components, the low frequent variations add a noise or error with high auto-correlation in time.

The sensor reacts to all, potentially high-frequent, changes in its environment. This local environment is unlikely isolated from the larger scale system. Thus, additional noise may be due to the sensor reacting to advected environmental properties instead of 'local' ones or due to the environment redistributing the sensor or the record. In the marine realm but also in lake domains, currents may influence the sensor, while in many domains the wind may affect the recording of the signal. Furthermore, small and large scale spatial variations of the process may affect the signal and contribute to the record. Our approach regards these contributions as noise. All these influences may introduce spatial and, here considered to be of more

importance, temporal correlations in those environmental properties, which we here consider as part of the noise term. We assume that advection from other regions by currents and wind, are especially important in contributing autocorrelation to our noise process. One can see these non-local factors as noise in the archive rather than the sensor.

Besides simple noise, redistributions of the environmental signal may also introduce biases in our estimate of the environment. Any bias is likely not fully time-constant but evolves with the environment on interannual, multi-decadal, and multi-centennial to millennial time-scales. The different time-scales result from the different time-scales of the environment. This is relevant for recent climate changes and interannual to interdecadal climate variability, but it becomes even more important for multi-millennial time-scales where we have to deal with the effects of changing seasons, glaciation, deglaciation, changes in bathymetry, and lithospheric adjustments. All of these processes may lead to biases, and such biases also lead to autocorrelation in the error.

One example of such time-evolving biases are changes in the seasonality of the environmental sensor. While one can see this again as a source of uncertainty in a narrowly defined proxy-system from sensor to reconstruction, it is in the end a bias of our attribution of the measurement to one season. We consider this bias on the sensor level. There are other potentially erroneous attributions besides the processes' seasonality. These are the location of the process in all three dimensions, for example, the habitat of living organisms, and a generally only partially correct calibration relationship. Again, these are environmental factors influencing the sensor and we consider them as noise here. However, they reflect a non-stationarity of our reconstruction-calibration-relation. Nevertheless, the idea that the modern relations between environment and proxy system worked over the full period of interest is a fundamental assumption of paleo-climatology (e.g., Bradley, 2015).

In the following, we assume three components to be important disturbances of the signal at the sensor level: the environmental noise, the redistribution, and the attribution errors. We reduce the latter to the potential biases due to changes in the seasonality. Taking all three components the sensor-record becomes

$$S(E) \approx \widehat{S}(T) + \eta_{env} + \eta_{redistr} + \eta_{season} \tag{3}$$

where we for the moment replace $\eta_i$ by $\eta_{env}$. In the following, we reduce these three components to two terms in our modifications of the input data.

### 3.1.2 Noise

First, we assume that $\eta_i$ includes both the effects of environmental dependencies and of redistribution. That is, $\eta_i = \eta_{env} + \eta_{redistr}$. This is the first error term. This in fact implies that we should consider auto-correlated noise-processes. If we only modify the model-output and concentrate on one parameter $T$, for example, temperature data, our pseudoproxy at this point becomes,

$$P(x, y, t, T) = P_T = T + \eta_i \tag{4}$$

The current version of $\eta_i$ is only a weakly correlated autoregressive (AR) process of order one, which we additionally scale by an ad hoc scaling factor. It thereby only includes a small part of the potential correlations among errors due to redistribution

and other processes. The innovations are sampled dependent on time and climate background from $\mathcal{N}(0, S(t)^2)$, where $S(t)$ is a time-dependent standard deviation. The time-dependence mimics a dependence of the noise on the background climate variability on long time-scales. Here, we use a 1000 year moving standard deviation $S(t_i) = \sigma(T(t_{i-499} : t_{i+500}))$. Our general formulation assumes that noise variability increases with increasing variability in the parameter $T$. Obviously, it could also be that noise variability reduces or reacts totally differently relative to the variability of $T$. The code includes a switch to invert the moving standard deviation about its mean or to randomize the orientation.

### 3.1.3 Bias

We can consider the changes of the seasonality, $\eta_{season}$, as an orbitally influenced bias term. We compute it for any latitude of interest. We apply the orbital bias term as additive but one may see it as a multiplicative or a nonlinear effect in many cases. Therefore the code uses it after the noise term $\eta_i$. The bias is the second error term in our formulation of modifications at the sensor level. The bias term is a scaling of the changes in annual latitudinal insolation but it is possible to choose different sub-annual time-periods of interest. The scaling is arbitrary and we refer to the provided code for details. The bias is zero in the year 0 BP. We set it to be positive if the insolation is larger; this can be randomized in the code. The amplitude of the bias is scaled by an ad hoc constant. The bias becomes notable at some latitudes but may be rather negligible elsewhere. We take the bias as $Bias(t) = \beta \cdot I_n$. Where $\beta$ is the scaling constant, and $I_n$ is a normalised and shifted insolation. $I_n$ is calculated as $I_n = ((I - \bar{I})/\sigma(I) \cdot q_{0.25} - I(t = 0BP) + 1)^u - 1$ for a chosen period. The chosen time-period influences the statistics included in the scaling. We consider the insolation since 150,000 BP. $q_{0.25}$ is the 25th percentile of the insolation data, u is generally 1, but can be sampled from $U = \{-1, 1\}$.

The pseudoproxy becomes

$$P_T(t) = T(t) + \eta_i(t) + Bias(t) \tag{5}$$

### 3.1.4 Results

Figure 2a shows an example of the initial noise $\eta_i$. The dependence on the background state is clearly visible for the visualized grid point data. There is an increase during the deglaciation and a multi-millennial reduction over the Holocene. Indeed, Rehfeld et al. (2018) diagnose a reduction in temperature variability from the Last Glacial Maximum to the Holocene by studying centennial to millennial time-scales.

Panel b of Figure 2 compares three potential amplitudes of the orbitally induced bias. We use the version with the smallest amplitude. Panel c of the Figure presents the grid-point temperature of the TraCE-21ka simulation and a simple 501-year running mean. The comparison with Figure 2d highlights that the effect of the bias is rather small given our choice of its amplitude. Nevertheless, comparing the panels also clarifies that our implementation of the bias results in a colder annual record over most of the considered time period while the record becomes slightly warmer in the very early portion of the simulated data.

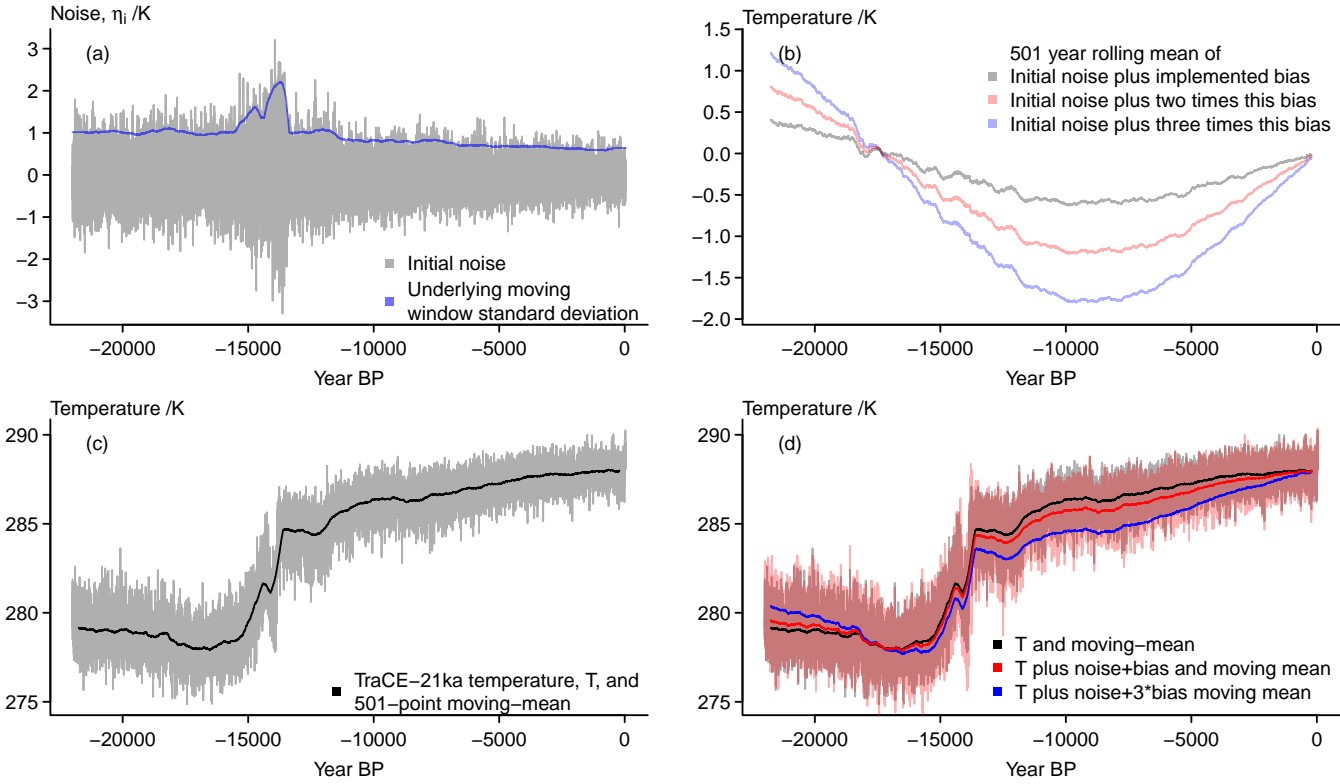

**Figure 2.** Visualising considered error sources at the sensor-stage: a) the initial noise and the underlying moving window standard deviations of the input signal, b) three versions of a potential bias as function of the local insolation, c) the input data and its 501-point moving mean, d) the input data and its 501-point moving mean plus noise and bias. The unsmoothed initial temperature is effectively hidden behind the unsmoothed temperature plus bias.

## 3.2 Assumptions on essential error sources 2: Archive

So far our approach describes a record of an environmental influence plus two error terms. This record becomes subsequently integrated in an archive. Afterwards, various processes may modify the archive or redistribute it. Modifications include selective destruction of parts of the record by processes acting all the time or by sparse random events or continually acting random

5 processes. Examples are bioturbation or re-suspension. These processes may result either in a correlated noise in time and space or simply white noise. Other de facto white noise errors may result from our finite and random sampling of the archive. However, this may be rather part of the observational noise.

Such modifications of the archive and sampling issues represent an important step in using inverse reconstruction methods because it is a priori not clear how the archive is generated and whether an individual measurement represents mean environ-

10 mental states or relates to single events. In this context, forward models and pseudoproxy approaches of sedimentary proxies

are a crucial tool in disentangling the controlling climatic environmental factors in the generation of sediment cores and their interpretation.

### 3.2.1 Smoothing and noise

Because we focus on sedimentary proxies, we argue that the archiving process foremost is a filter of variability above a certain frequency level, for example, by diffusive processes or bioturbation (compare Dolman and Laepple, 2018, and their references). Dependent on the system in question this may only affect the very high frequencies but for other systems it may extend to multi-decadal or even centennial to millennial frequencies. On top of this smoothing of the archive, there may be additional noise as the smoothing function is unlikely homogeneous. We assume such a filtering to be the fundamental modification of the record in the archive, and, thus, only consider this process in our archive modelling.

Inspired by the simple proxy forward formulations of Laepple and Huybers (2013, see also Dolman and Laepple, 2018), we produce five different versions of the archived pseudoproxy-series. The first and second series are simple running averages of the sensor record on which we add a highly autocorrelated AR-process of order one. The two versions differ in the length of the averaging window, the AR-coefficients, and the standard-deviations of the innovations. The versions three and four similarly differ in the amount of average smoothing, but we use random window lengths for each date. The rationale for the two different smoothing lengths is to represent both strongly and only slightly smoothed proxies.

The fifth version aims to mimic the behavior of proxies when researchers use only a small part of an available proxy, e.g., pick only a certain number of samples. An example is the simple forward formulation for Mg/Ca proxies by Laepple and Huybers (2013, see also Dolman and Laepple, 2018).

Smoothing lengths and random factors in this approach could depend on the background climate. Indeed, the code includes options for the random smoothing lengths to depend on the mean climate or the climate variability. The provided data uses an approach where the random smoothing lengths follow an autoregressive process around a climate dependent reference smoothing length, where, considering Vardaro et al. (2009), warmer climates result in shorter smoothing intervals. The smoothed archive records are then either

$$P_T(t) = g_1(T(t) + \eta_i(t) + Bias(t), t) \tag{6}$$

where $g_1(t)$ is the time dependent filter, or

$$P_T(t) = g_2(T(t) + \eta_i(t) + Bias(t)) + AR \tag{7}$$

where $g_2$ is the constant smoothing and we add an AR-process to account for the inhomogeneities in the smoothing.

The fifth version of the pseudoproxies subsamples over the random filter interval and adds a noise term to mimic a seasonal uncertainty. That is, we sample $n$ years within the filter interval, and take the mean over the temperature and the noise for these years. We add another noise term to represent the intra-annual seasonal uncertainty. $P_T$ in this case becomes

$$P_T = h(T(t), t) + h(\eta_i(t), t) + \eta_s \tag{8}$$

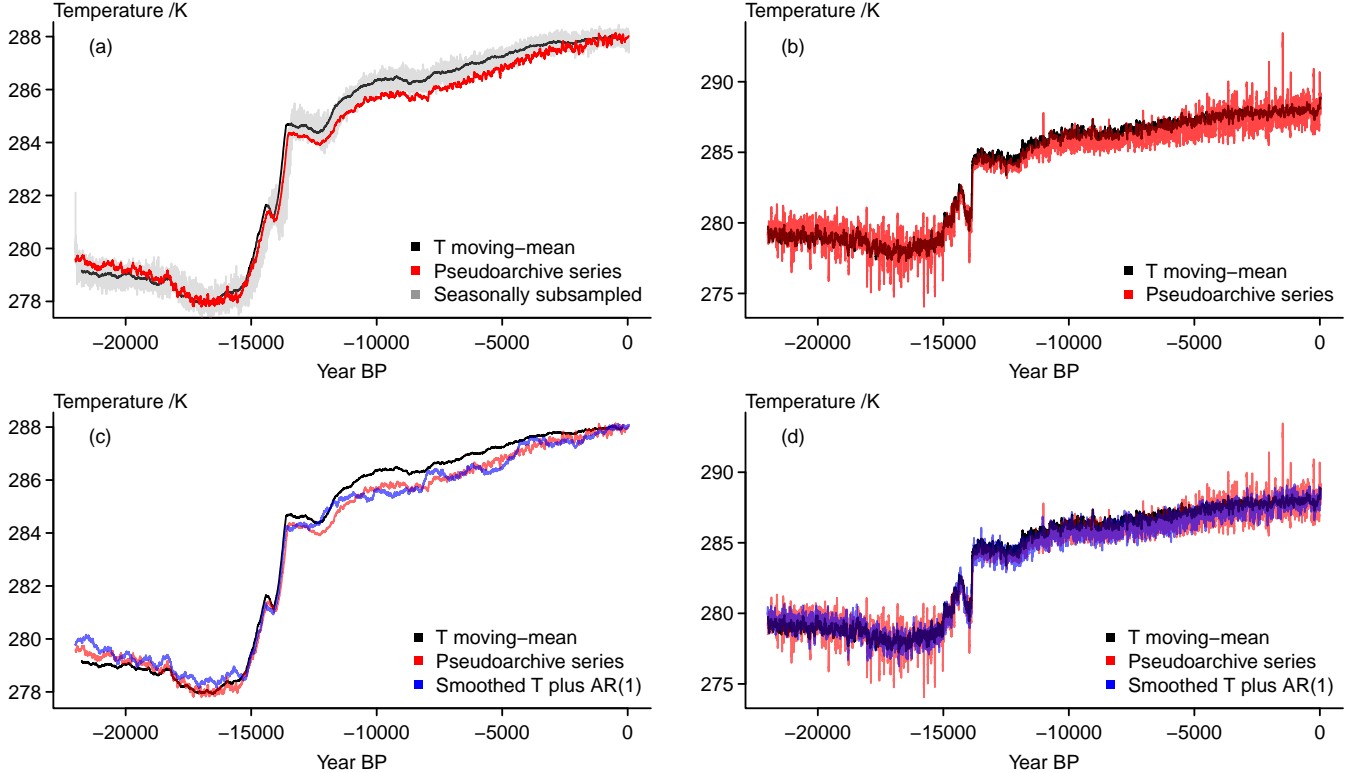

**Figure 3.** Visualising considered error sources at the archive-stage: a) 501-year moving mean of the input data, the pseudo-archive series with longer average smoothing lengths, and the subsampled record, b) 501-year moving mean of the input data, the pseudo-archive series with shorter average smoothing lengths, c) 501-year moving mean of the input data, the pseudo-archive series with longer average smoothing lengths, and the version with constant smoothing and added AR(1)-process, d) 501-year moving mean of the input data, the pseudo-archive series with shorter average smoothing lengths, and the version with shorter constant smoothing and added AR(1)-process.

where $h(t)$ represents the sub-sampling and $\eta_s$ the intra-annual noise. We do not include the bias term for the subsampled proxies. On the one hand we apply the bias only for the mean annual temperature, i.e. individual seasons show different biases. While we could account for this by sampling the biases over the different seasons or even months in producing $h(t)$ or $\eta_s$, we prefer to keep our model simpler. Excluding the bias term may be interpreted as the seasonal subsampling cancelling out

5   the bias. In reality any cancellation would not result in a convergence on the simulated climate state but more likely on a recorded value between the biased and the 'true' climate. The coded version of the sub-sampling still includes the bias-term as a comment.

### 3.2.2   Results

The biased moving average already shows the differences between the target temperature and the pseudoproxy-record (compare

10   Figure 2. The pseudo-archive-series in Figure 3a shows this more clearly. Here we use a randomized smoothing interval.

Differences are less visible for shorter random smoothing intervals (compare Figure 3b). Further panels of Figure 3 add the constant smoothing archive approximations which we modify by an additional highly correlated AR-process (Figure 3c and d). This procedure randomly amplifies, dampens, or inverts certain biases in the presented case. That is, while the simple random smoothing may emphasize the bias, the AR-procedure overlies this bias with additional variations.

The panels highlight an apparent offset between the randomly smoothed archive series, the constantly smoothed archive series, and the smoothed input data. The smoothed version of the input data as well as the constant filtering use a centered approach, that is they are symmetric about their date. The time varying smoothing tries more realistically to imitate a bioturbation approach (compare Dolman and Laepple, 2018, and their references) and thus provides a shift in the series.

Figure 3a also shows the seasonally subsampled pseudo-archive-proxy. The data ignores the bias term and the resulting
series is by construction symmetric around the original data, our target. Nevertheless, there are pronounced deviations from the original data. Considering only the deviations from the target temperature moving mean highlights that this approach is notably more noisy than the filtered data but preserves pronounced longer term excursions of the input data (not shown).

### 3.3   Assumptions on essential error sources 3: Measurements

The archiving represents also a transformation from time-units to archive-distance-units, to depths, rings, distances. The proxy
becomes a tuple of date and data. Now the dates are uncertain as each data-point includes information from different original dates due to the smoothing function. The sampling may lead to additional uncertainties due to disturbances of the archive, and the dating of our samples is a profoundly uncertain process.

#### 3.3.1   Measurement error

Prior to dealing with errors due to dating uncertainty, we take an additional noise term to represent measurement errors and
apply this for each date to account for the potentially imperfectly measured series. The term includes not only the errors introduced by our assumed methods of measuring the proxies and the methods' potential to make mistakes. This "true" measurement error may result in biases due to limits of what our methods can detect or systematic offsets due to a laboratory-specific, potentially erroneous, approach to the measurement. Potential offsets imply that we should generally expect a certain amount of auto-correlation in this noise. The term has further to account for the accidental handling of the records in the laboratory, for
example, influences from storage or from other processing of the samples and the data, which may result in autocorrelated errors if these influences have a systematic component. Thus, it is not necessarily the case that we can consider inter-laboratory reproducibility as white noise. However, the intra-laboratory repeatability is likely indeed a white random process. We also assume repeatability and reproducibility to be part of our measurement error term. While we just mentioned various reasons to assume autocorrelation in this error-term, we only provide a white noise term for the measurement noise. Again, the code
allows to modify this.

We apply the measurement error term at the end. However, we introduce this term before dealing with the dating uncertainty since we provide proxies without dating uncertainty. The measured proxy-series becomes

$$M_T = P_T + \eta_M \tag{9}$$

In reality, we do not have a continuously sampled series, but obtain only samples at certain intervals. Assuming $N$ samples the sampled pseudoproxy becomes

$$P_{P_T} = P_T(t = \{t_1, ..., t_N\}) \tag{10}$$

The sampling of the archive likely produces errors in the samples. We assume these are included in the measurement uncertainty. We provide at each grid-point sampled series of the pseudoproxies detailed above. We do not distinguish between different sampling techniques. We simply sample the records at certain dates and add the described noise term.

### 3.3.2 Dating uncertainty

Dating uncertainty represents a big part of our overall uncertainty for many proxies, especially for sedimentary proxy-records. In our framework, already the smoothing function redistributes information from one date across the archive. Usually one considers this temporal uncertainty separately from the proxy-record error. For assessing reconstruction methods and simulations, it would be beneficial to be able to include dating uncertainty within the proxy-error. That is, if we consider proxies as tuples of data and date, we have to transform the uncertainty of the date into an error-term for the data. In the following we distinguish between the dating uncertainty, that is the uncertainty that a sample is from a certain date, and the dating error, by which we mean the potential error in our (pseudo)proxy due to the uncertain dating.

There are a number of approaches to transfer the dating uncertainty towards the proxy-record error (e.g., Breitenbach et al., 2012; Goswami et al., 2014; Boers et al., 2017). Ensemble and Bayesian age-depth modelling approaches also allow to infer an additional error term (e.g., Haslett and Parnell, 2008; Blaauw and Christen, 2011). However in the present application, we want to capture the error in a time-series. Thus, we take a very simple approach, which assumes that the error due to dating uncertainties is related to the climate state over the period of the dating uncertainty. Nevertheless, since we provide sample dates and random sampling uncertainties, the application of age modelling to the pseudoproxies is in principle possible (e.g., following the approach of Dee et al., 2015, 2018).

The code includes several variations of our estimation of an effective dating error. These reflect different amounts of dependence between subsequent samples. In all variants, we only consider dependence between two subsequent samples while for real proxies the correlations may extend across larger portions of the proxy-record. The following general approach is common to all variations of our procedure: First, we sample uncertainties in time for each sample date. We take these as dating uncertainty standard deviations. These uncertainties can be sampled fully randomly or dependent on the available smoothing interval data from the archive stage. Then we take the effective dating error at each sample date/depth to be a random sample from a normal distribution.

The mean of this distribution is the difference between the sample-data and the mean over the data within plus and minus two dating uncertainty standard deviations. The standard deviation of the distribution is the standard deviation of the differences

between the individual data points within this interval and this mean. The effective dating error is then

$$\epsilon_D = \mathcal{N}(\overline{P_{T_D}}, \sigma_D^2) \tag{11}$$

where

$$\overline{P_{T_D}} = \overline{P_T(t_S = \{t_{i-2\sigma_{dating}}, ..., t_i, ..., t_{i+2\sigma_{dating}}\})} - P_T(t = t_i) \tag{12}$$

is the mean over the region of influence and

$$\sigma_D^2 = E[(P_T(t_S) - \overline{P_{T_D}})^2] \tag{13}$$

is the variance of the distribution.

    In the simplest formulation ignoring the dependence between subsequent dates, the sampled pseudoproxies become

$$P_{P_T}(t_1, ..., t_N) = g(T(t) + \eta_i(t) + Bias(t), t)(t_1, ..., t_N) + \epsilon_D(t_1, ..., t_N) \tag{14}$$

Alternative formulations of the pseudoproxy become

$$P_{P_T}(t_1, ..., t_N) = g(T(t) + \eta_i(t) + Bias(t))(t_1, ..., t_N) + AR_i(t_1, ..., t_N) + \epsilon_D(t_1, ..., t_N) \tag{15}$$

or

$$P_{P_T}(t_1, ..., t_N) = h(T(t), t)(t_1, ..., t_N) + h(\eta_i(t), t)(t_1, ..., t_N) + \eta_s(t_1, ..., t_N) + \epsilon_D(t_1, ..., t_N) \tag{16}$$

This initial formulation of the effective dating uncertainty error ignores potential correlation between the dating errors. The most simple way to account for this makes subsequent errors dependent

$$\epsilon_{D_i} = \rho \cdot (\epsilon_{\xi_{D_{i-1}}} + (P_{P_{T_{i-1}}} - P_{P_{T_i}})) + \epsilon_{\xi_{D_i}} \tag{17}$$

This formulation has only a minor influence on the results. It is included in the code via a binary switch.

    A slightly more complex formulation makes the error term at each date dependent on the previous sample's age uncertainties
and mean data. Previous refers to archive units instead of time units. Then the dating error becomes

$$\epsilon_{D_i} = \rho \cdot (\epsilon_{D_{i-1}} + (P_{P_{T_{i-1}}} - P_{P_{T_i}})) + \epsilon_{\xi_{D_i}} \tag{18}$$

where $\epsilon_{\xi_{D_i}}$ are the random innovations for date $i$. Our initial choice of $\rho = 0.9$ can give large effective dating uncertainty errors. A switch in the code allows to use this inter-dependent error. Another switch allows to consider the dependence between samples as a function of their dates and the dating uncertainty,

$\rho(t) = 1 - (t_i - t_{i-1})/(2 \cdot \sigma_d(i-1)) \tag{19}$

The time-dependent dating uncertainty for each date $\sigma_d(t)$ is generated randomly (compare above $\sigma_D$). We provide data for the case with a time-dependent $\rho(t)$.

Alternative simple formulations may include different noise processes like noise generated from Gamma-distributions. The available smoothing interval data can inform the sampled dating uncertainty. We could further use this information to provide a deterministic, not random, error for each sampled date, that is we could take a bias based on all dates influencing the selected date within the dating uncertainty.

In our current setup the age uncertainty does not depend on the measurement noise. The measurement error is added afterwards to the series including the effective dating uncertainty error. This decision is arbitrary. On the one hand a classical dating uncertainty affects the measured value. Then, also $P_{P_T}$ above should already include the measurement error. On the other hand, the dating uncertainty affects the archived values independent of the measurement noise. Therefore we keep both independent.

The measured proxy-series becomes

$$M_T = P_{P_T} + \eta_M \tag{20}$$

The final proxy is in temperature units as is the initial input data. We ignore a separate term for potentially non-linear and climate-state dependent errors in our calibration relationship and assume the measurement noise term accounts for this as well. A separate term could be again a state-dependent Gaussian noise. It could also be a noise from a skewed distribution whose mode depends on the background climate. On the other hand, a state-dependent bias term could simulate a mis-specified calibration relation while a time-dependent bias term could simulate a degenerative effect over time within the archived series. None of these are included in the current version.

### 3.3.3 Results

Figure 4 shows versions of an archived proxy plus interannual measurement noise. The panels give an impression of how a proxy would look from measurements on a perfectly annually sampled archive. The final amplitude of the noisy proxy is generally slightly smaller for all versions of our pseudoproxies than the amplitude of the interannual variations for the chosen location. This may be different at other locations. The different versions of the smoothing and of the smoothing plus AR approaches are shown in Figure 4a and b, respectively. Figure 4c plots the seasonally subsampled pseudoproxy. The final versions of the pseudoproxies generally preserve previously included biases.

Figure 5 presents a number of series sampled at $N = 200$ dates. All panels include the original temperature data sampled at these 200 dates. The Figure emphasizes how the initial temperature variability at the chosen grid-point is generally slightly larger than any of our uncertainty estimates. Our effective dating uncertainty error seldom results in large deviations from the archived record. The subsequently applied measurement error also only seldom leads to large offsets compared to either the original data or the effectively date uncertain record. Thus, for our chosen parameter settings and the shown grid-point, the pseudoproxies fall within the range of the initial estimates. In turn, if we assume we have reliable calibration relationships, our calibrated proxy-series should also be reliable estimates of the past states.

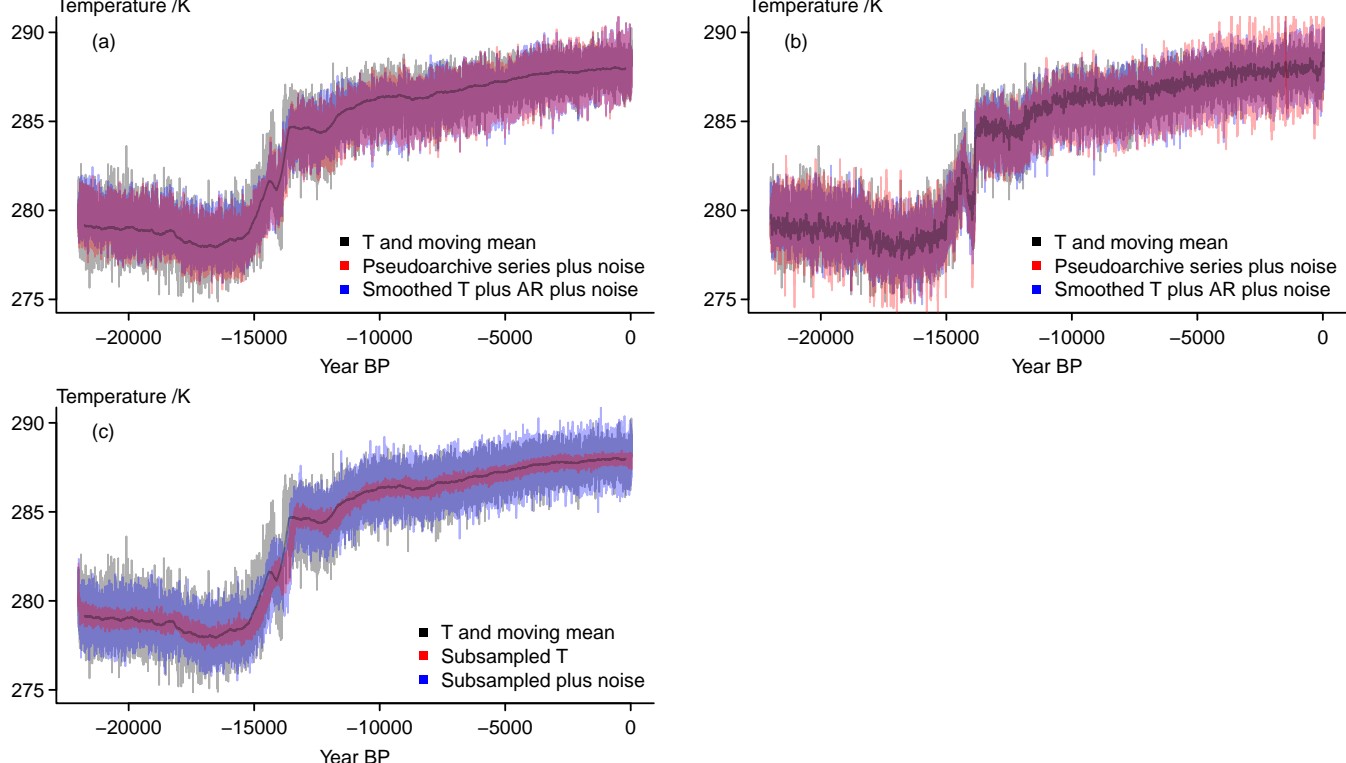

**Figure 4.** Visualising considered error sources at the measurement-stage for the full series: a) 501-year moving mean of the input data, the pseudo-archive series with longer average smoothing lengths and the constant smoothing plus AR series with added measurement noise, b) 501-year moving mean of the input data, the pseudo-archive series with shorter average smoothing lengths and the constant smoothing plus AR series with added measurement noise, c) 501-year moving mean of the input data, the subsampled record, and the subsampled record with added measurement noise.

Nevertheless, the biased estimates occasionally are only bad matches for the original data. This is also the case for the subsampled data where we did not include the bias. Comparing the sampled pseudoproxy series to the smoothed original temperature data (compare Figure 5a) highlights that estimates for past climates may well fall within the range of the original interannual temperature variability but may nevertheless strongly misrepresent the mean climate represented by the sample.

Considering the effective dating uncertainty error, the discrepancies between input data and pseudoproxy are rather small for uncorrelated or weakly correlated age uncertainties. However, in the case of strong dependencies between subsequent data, pronounced biases and mismatches may occur (not shown). The assumed co-relation between two dates has a strong influence on the size of these mismatches. We show the case for a time-dependent co-relation between subsequent dates, which gives intermediately sized mismatches.

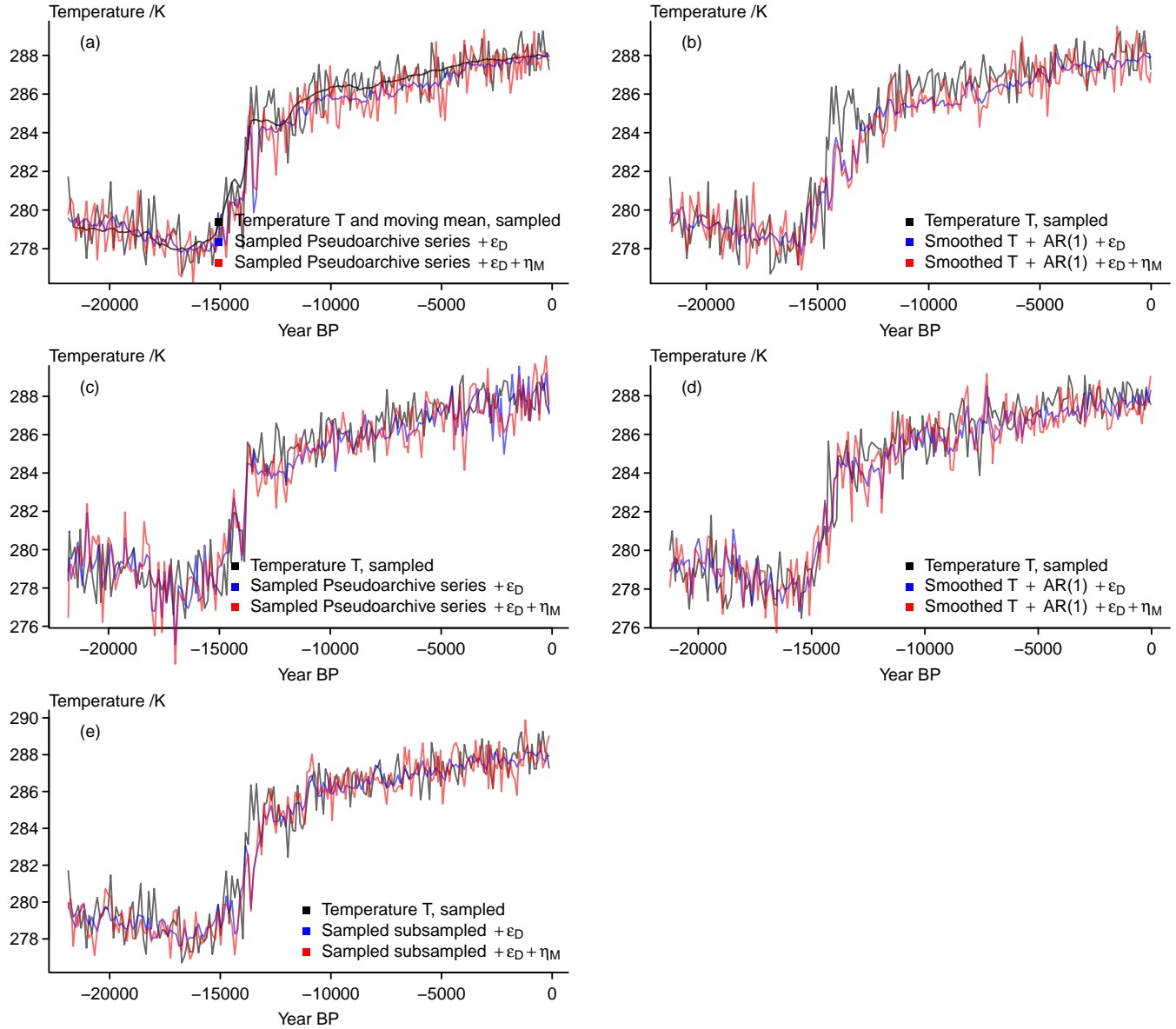

**Figure 5.** Visualising the sampled records: a) Input data and its 501-year moving mean, the pseudo-archive series with longer average smoothing lengths plus the effective dating error and plus the effective dating error and measurement noise, b) input data and its 501-year moving mean, the constantly smoothed record with longer smoothing length plus AR series with added effective dating error and with added effective dating error and measurement noise, c) input data and its 501-year moving mean, the pseudo-archive series with shorter average smoothing lengths plus the effective dating error and plus the effective dating error and measurement noise, d) input data and its 501-year moving mean, the constantly smoothed record with shorter smoothing length plus AR series with added effective dating error and with added effective dating error and measurement noise, e) input data and its 501-year moving mean, the subsampled record with added effective dating error and with added effective dating error and measurement noise.

### 3.4 General Results

Figures 2 to 4 present the different versions of the pseudoproxies for the chosen location. Under our assumptions, the influence of the orbital bias term is notable. The approaches using time-dependent smoothing or simple smoothing plus an AR-process may nearly or fully cancel the bias. This effect is less prominent for the time-dependent filter. Generally, both approaches seem to have similar effects.

Figure 4 includes the effect when we hypothetically add measurement noise at every date. Under our assumptions this noise is still smaller than or only as large as the original interannual variability but, including biases, mean estimates may be outside of the interannual variability of the original data. In these examples, the variability of the subsampled proxies is comparable to the smoothed ones after a measurement error is added. It is interesting to note that for the smaller smoothing the AR-process seems to cancel the orbital bias more strongly in Figure 3. Figure 5 shows the data-sets sampled at $N = 200$ dates. It clarifies the error described for the interannual data. The document assets provide equivalent visualisations for another grid-point. These generally confirm the above descriptions.

#### 3.4.1 Spectral power

Figure 6 adds a comparison of power spectral densities computed from a wavelet based approach similar to the Weighted Wavelet Z-transform of Foster (1996). The approach is described by Mathias et al. (2004) and McKay and colleagues provide a compiled version at https://github.com/nickmckay/nuspectral (last accessed, 11 March 2019) (Nick McKay et al.). Due to the length of computation, we do not show the density for the full 22,040 year input data but only for a record sampled every ten years. Results may be specific for the chosen grid-point.

The Figure shows estimates for the full records and for the data of the last twelve thousand years of the records. Spectral densities for the regularly sampled original temperature data in Figure 6a highlight that the differentiation between full and late records results in prominent differences for multi-centennial to millennial periods. On the other hand, differences are smaller for the irregularly sampled input temperature data but still notable for millennial periods. However, there is an offset between the irregularly sampled data and the regularly sampled input data.

Spectra for full and late records of the various pseudoproxies are generally similar to the irregularly sampled input data spectra (Figure 6b-f) but the offset to the input data can be smaller than in Figure 6a. Differences between sampled late and full records are often largest at intermediate millennial periods. Deviations are largest for the subsampled pseudoproxy approach at long periods (Figure 6f) but they become also notable for the constant smoothing approaches at shorter periods in the centennial band (Figure 6c,e). This is mainly due to the characteristics of the full period spectra for the constant smoothing, which show an increase in power spectral density for shorter and longer periods. That is, the constant smoothing full period spectra remind of grey noise spectra. Despite these differences and the apparent offset to the input data spectra, the irregularly sampled spectra for all cases are rather similar.

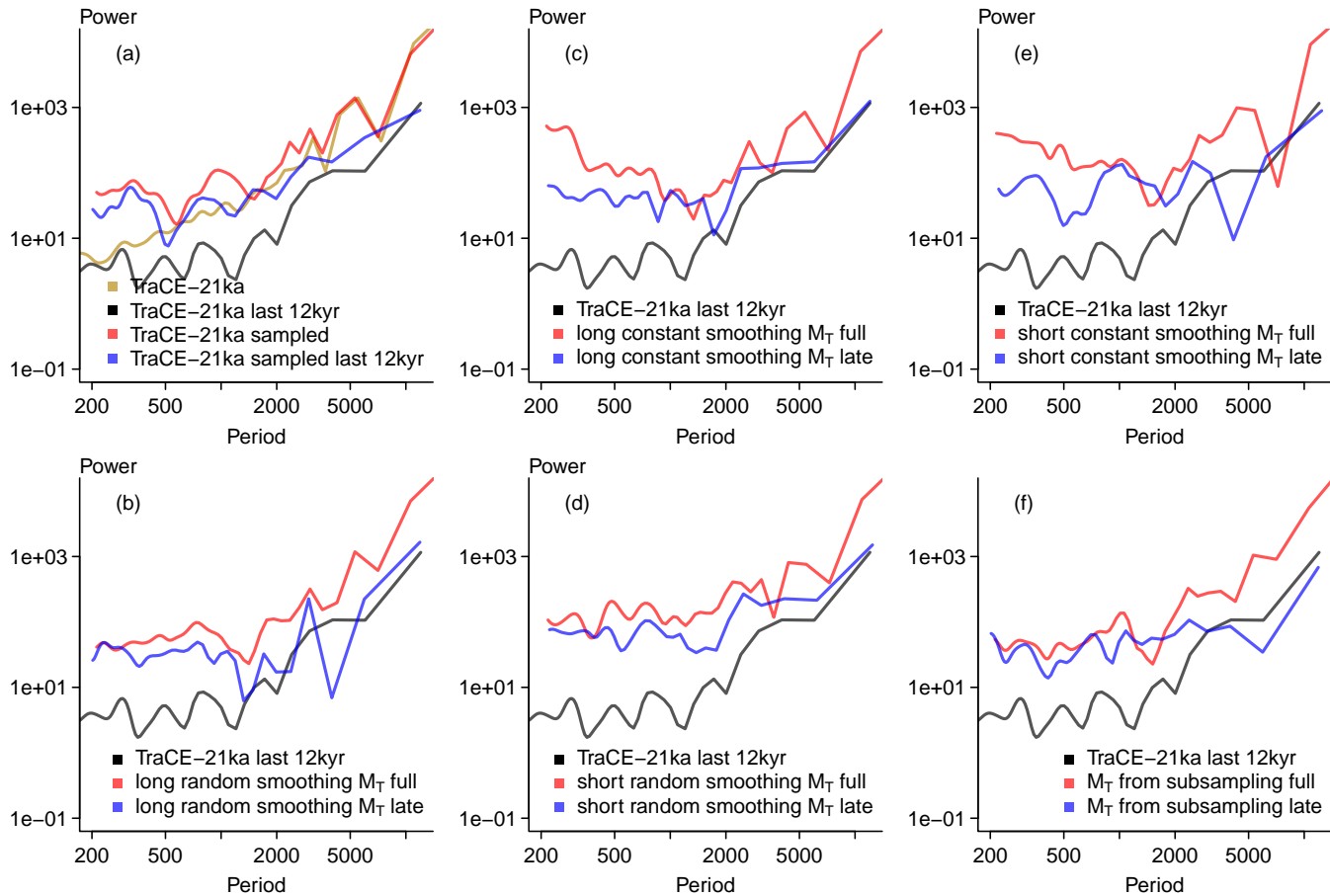

**Figure 6.** Wavelet based power spectral densities (Mathias et al., 2004; Nick McKay et al.). Densities are weighted following Mathias et al. (2004) to smooth the records for ease of comparison. Lines are for records split up by first 10k years of the records and the last 12k years of the records. Input data refers to the input data at 10 year intervals. All panels include the late input data from the TraCE-21ka simulation as black lines, red lines are in all panels for a full period record, blue lines are in all panels for the last 12k years of the version of a pseudoproxy. In addition to the input data from the TraCE-21ka simulation the panels show: a) the sampled TraCE-21ka simulation input data, b) the sampled pseudoarchive-series with long average smoothing plus the effective dating error and the measurement noise (long random smoothing $M_T$), c) the constantly smoothed record with a longer smoothing plus an AR(1)-process and including the effective dating error and the measurement noise (long constant smoothing $M_T$), d) the sampled pseudoarchive-series with short average smoothing plus the effective dating error and the measurement noise (short random smoothing $M_T$), e) the constantly smoothed record with a shorter smoothing plus an AR(1)-process and including the effective dating error and the measurement noise (short constant smoothing $M_T$), f) the subsampled data plus the effective dating error and the measurement noise ($M_T$ from subsampling).

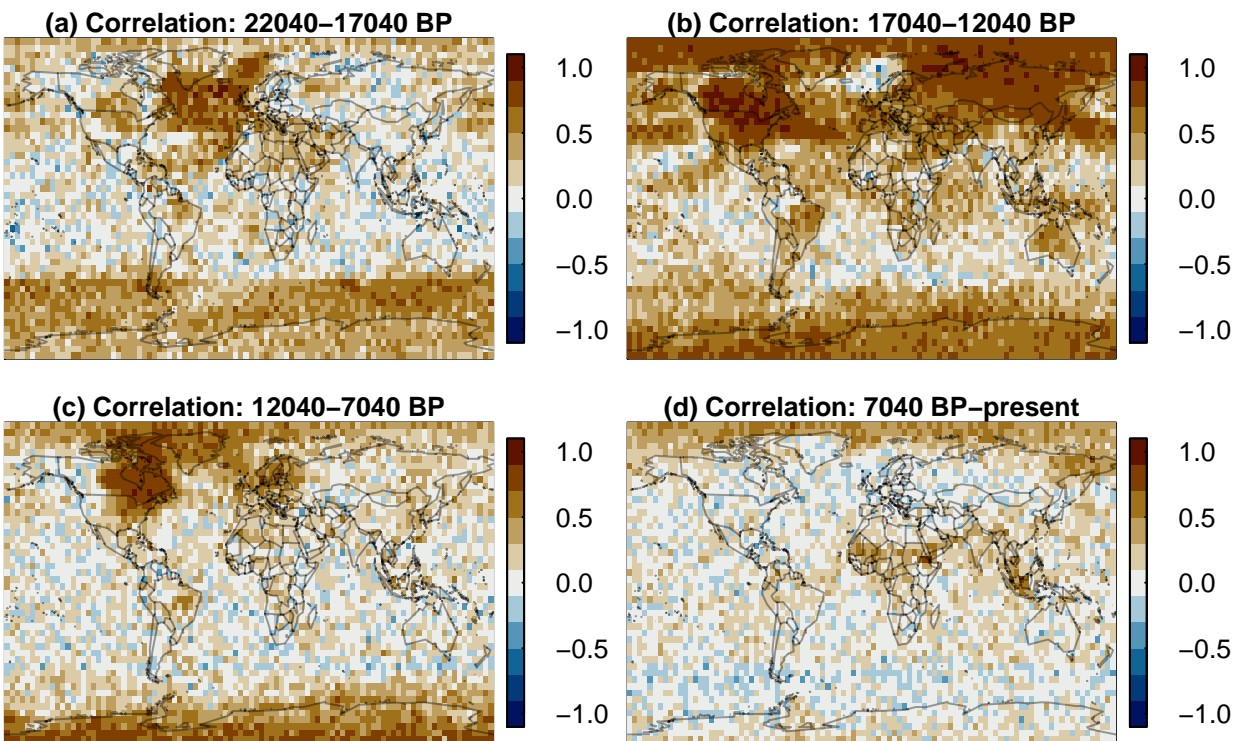

**Figure 7.** Point by point correlation maps between input data and the smoothed record plus AR(1)-process plus effective dating error and measurement noise for the sample dates within the first (a), second (b) and third (c) subsequent 5,000 year windows of the record and the samples within the remaining years (d).

### 3.4.2 Global data

The supplementary assets for this manuscript include plots of selected series from our analyses at all grid-points starting from the south towards the north (Supplementary document 1 Figure 1 at https://doi.org/10.17605/OSF.IO/ZBEHX/). These series are the input data at the grid-point, the smoothed-plus-AR-process series at the grid-point, and its subsampled version including all uncertainties.

These plots highlight three main points. First, peaks and troughs at some location are clearly attributable to the specific implementation of the forcing in the TraCE-21ka simulation (He, 2011; see also Liu et al., 2009). That is, these signals are not realistic but due to technical decisions in the production of the simulations. Furthermore there is potentially unrealistic variability at some grid-points for some periods. Second, the bias term in its current version may have only a small influence at certain latitudes. Third, our noise model shows often larger effects in the mid latitudes and the tropics. There is also a longitudinal dependence. Supplementary document 2 Figure 1 (https://doi.org/10.17605/OSF.IO/ZBEHX/) emphasizes the regional differences in the long term climate evolutions by selecting only grid-points in equal intervals to provide a more intuitive view of the globe. Similarly, Supplementary document 2 Figure 2 adds scatter plots of the pseudoproxy on the y-axis

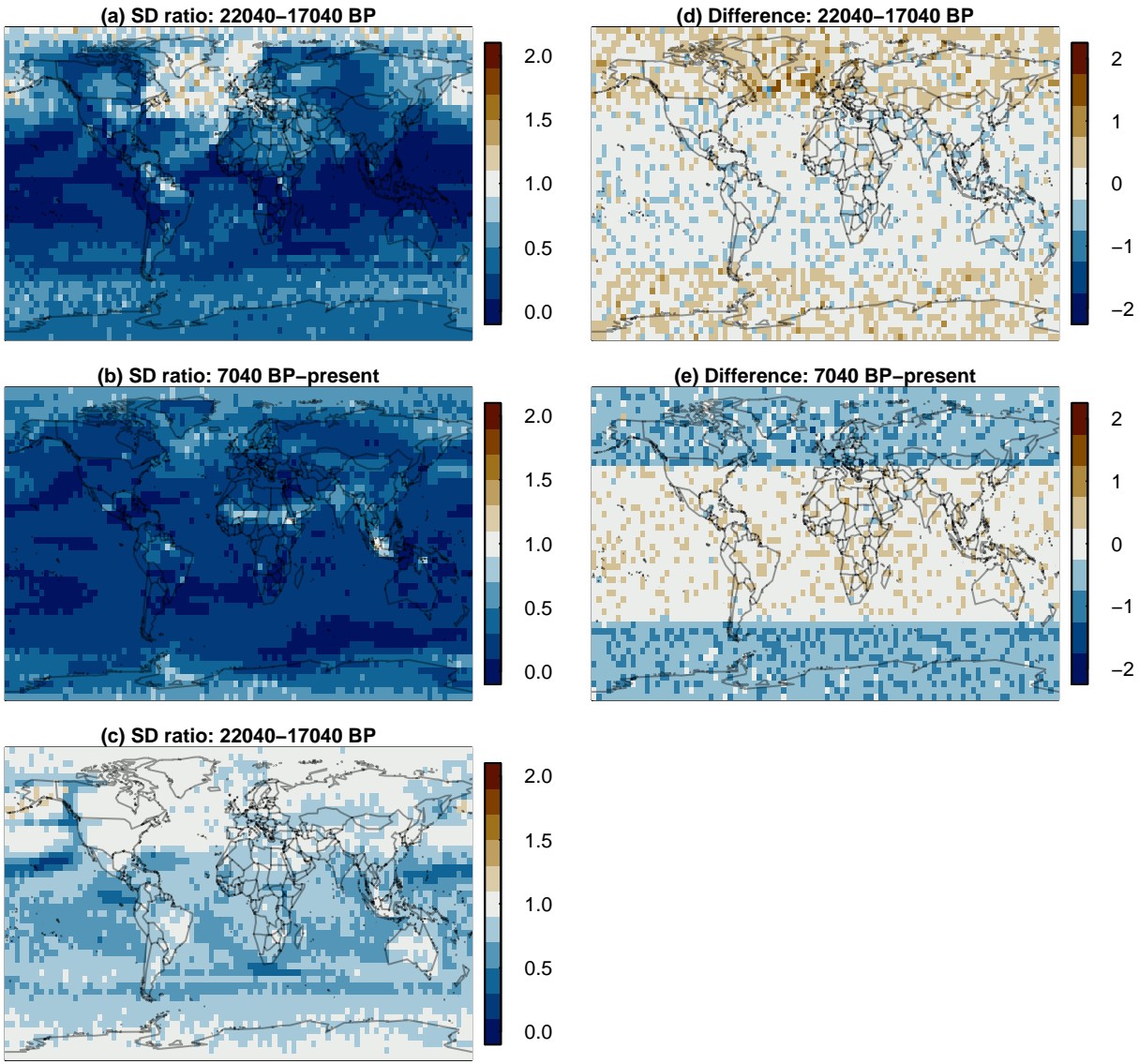

**Figure 8.** Left, standard deviation ratios of the sampled sampled 501-year moving mean input data relative to the smoothed record plus AR(1)-process and the effective dating error and the measurement noise for the samples in the first 5,000 years of record (a), the last 7,040 years of the record (b), and the full record (c). Right, differences between the mean of the sampled input data and the mean of the smoothed record plus AR(1)-process and the effective dating error and the measurement noise for the samples in the first 5,000 years of record (d) and the last 7040 years of the record (e).

against the original data on the x-axis for a small selection of grid-points, highlighting the common lack of a clear relation besides the deglaciation.

Figure 7 provides correlation coefficients between the sampled interannual grid-point data and the pseudoproxies including all uncertainties for the strong smoothing plus AR. The four panels show correlations for those samples within the first, second, and third 5,000 year chunks of the original data, and those samples in the remaining years. We choose to present the data this way to avoid detrending the data over the deglaciation interval. Relations between original data and pseudoproxies are generally weakest in the tropical belt. In the period until present, correlations are overall weak. High latitude correlations are most notable during the deglaciation and slightly less notable during the first millennia of the Holocene. In these periods, correlations appear to be largest in areas with glacial remnants.

Figure 8 adds for the first, the last, and the full period the relative standard-deviation $\sigma_{T21k}/\sigma_P$ in the left column and the bias $\bar{T}_{T21k} - \bar{T}_P$ in the right column. T21k refers to the simulation, P to the pseudoproxies. For the standard deviation ratios, we use 501-year moving averages of the TraCE-21ka data. Variability is generally larger in the pseudoproxies except for the North Atlantic and the northern high latitudes in the early period, and it is larger in the pseudoproxies more or less everywhere in the late period. Over the full period, variability is notably larger mainly in the tropics and the southern hemisphere, it is about equal over Antarctica and wide regions of the northern Hemisphere. The variability is clearly larger in the input data only over a small region in the northern Pacific.

The overall largest bias occurs off the coast of southeastern Greenland in the early period in Figure 8. Otherwise there is a spatial separation between the mid- to high latitudes and the tropics and subtropics for both periods. The bias is more prominent in the higher latitudes where it is predominantly positive in the early period but predominantly negative in the late period. Obviously, the general latitudinal bias pattern is by construction because we construct the bias as function of latitudinal insolation.

## 3.5 On generalizations of the errors

While we already chose comparatively simple procedures for our approach to obtain pseudoproxies from a model simulation, it is likely possible to simplify these to a higher degree. Such a general expression for the error in proxies over multi-millennial time-scales may be more usable in a number of ad-hoc model evaluations and model-data comparisons. Most importantly, such a generalized approach also allows to quickly produce ensembles of pseudoproxies.

Following our previous assumptions, the easiest way to obtain such a generalized error-model would be to assume a simple, potentially correlated noise model for the sensitivity of the sensor to the environment. Here, we use an AR-process of order one with AR-coefficient $\phi = 0.7$. Either here or later one scales the series or adds a bias term to account for changing seasonality over multi-millennial time-scales. The sum of the input data and this error are then subject to a simple moving averaging function. On top of this another simple correlated noise process mimics that the redistribution in the archive is not constant in time. Another random component accounts for the measurement error. Thus, simple correlated noise may be enough to catch

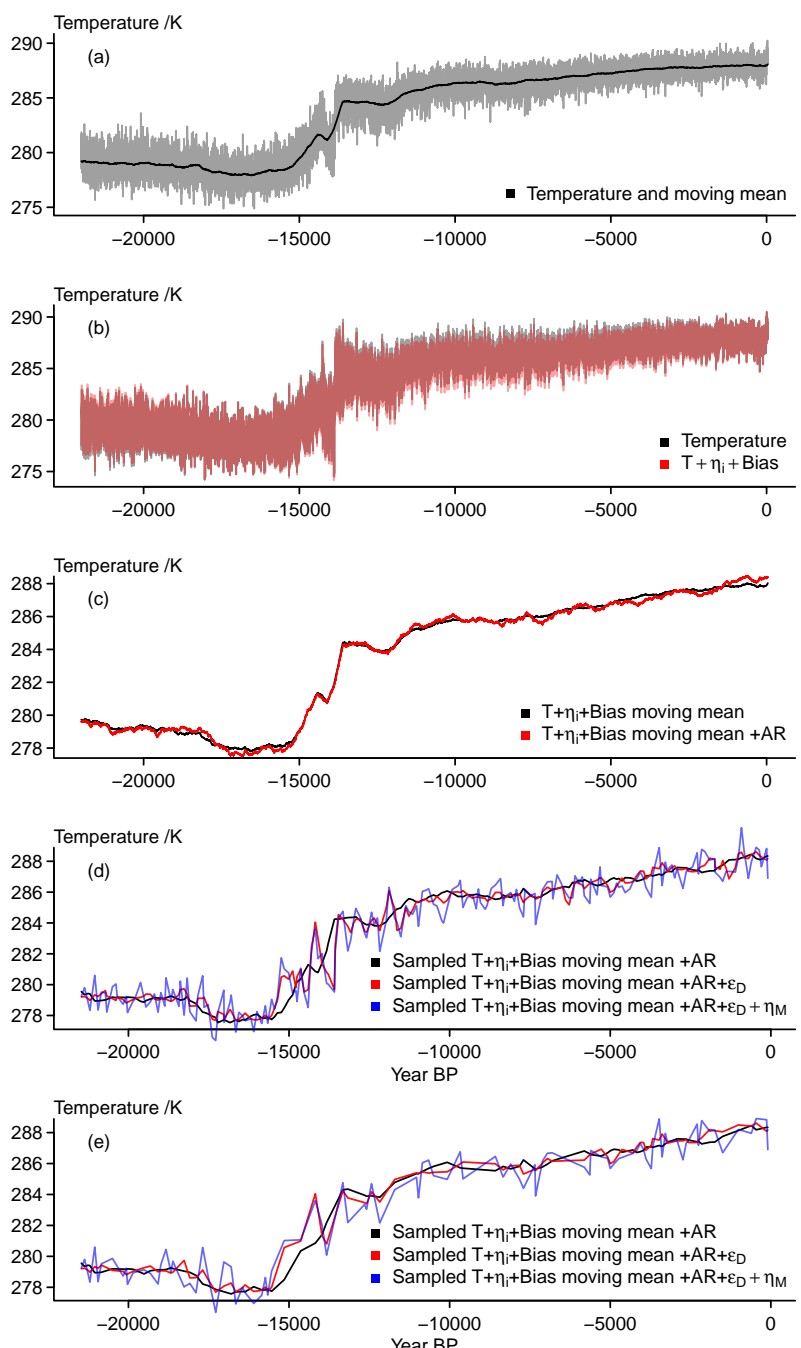

**Figure 9.** Visualising the simplified essence of the surrogate proxy calculations: a) input data and 501-point moving mean, b) input data plus initial noise and bias term, c) moving mean of input data plus noise plus bias and the same record plus an AR(1)-process, d) smoothed temperature plus noise plus bias plus AR-process sampled at 200 dates, this record plus the effective dating error, and this record plus the effective dating error and measurement noise, e) smoothed temperature plus noise plus bias plus AR-process sampled at 100 dates, this record plus the effective dating error, and this record plus the effective dating error and measurement noise.

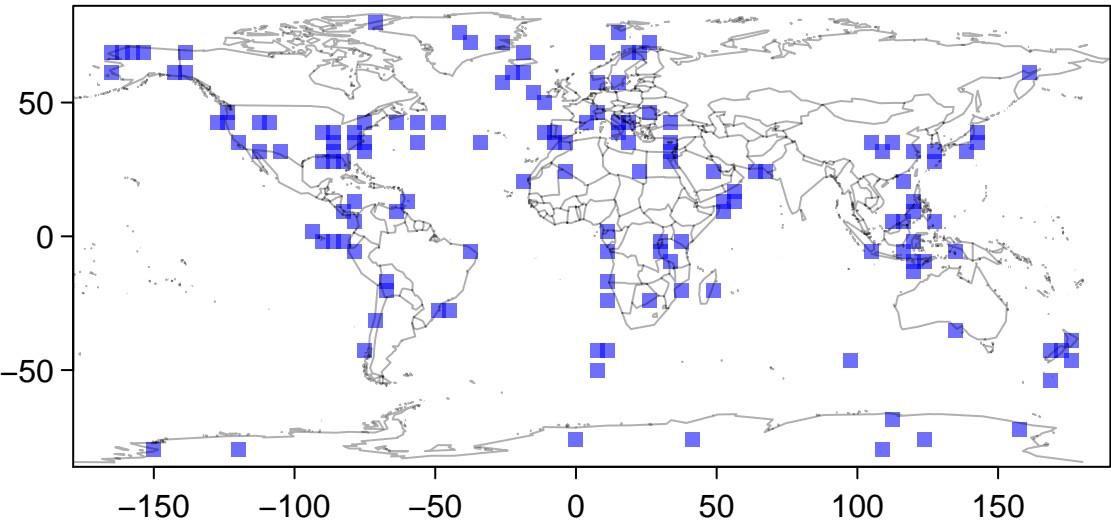

**Figure 10.** Map of the locations for the ensemble of surrogate proxies.

the essence of the error. In short, the generalized pseudoproxy becomes:

$$M_T(t_1,...,t_N) = g(T(t) + \eta_i t + Bias(t)) + \epsilon_D(t_1,...,t_N) + \eta_M \tag{21}$$

where $g$ is the smoothing, $\eta_i$ is the initial noise, $Bias$ is the bias term, $\epsilon_D$ is the effective dating error, and $\eta_M$ is the measurement error. This is conceptually identical to the smoothing plus AR approach presented above. Its derivation is less grounded in real proxies. The provided data differs only in the amount of autocorrelation in the noise terms.

Figure 9 summarises results for the generalized approach. It clarifies that while an error may mask certain features of the past climate evolution, this simple generalized pseudoproxy-generation is unlikely to distort the proxy completely if we take the assumptions made above to be approximately appropriate. Interestingly, the generalization appears to modify the input signal slightly less than the more complex approach. However, as we display slightly different data comparisons here, it is more appropriate to note that the dating uncertainty has only a minor effect compared to the initial bias and AR-process modifications and compared to the subsequent addition of the measurement noise.

While researchers may validly wish for such simplified recipes for producing pseudoproxies, using a full or at least more complex process-based approach is advisable, if it is necessary to account for effects of biology, environmental long-term changes, and other weakly constrained uncertainties. More complex approaches further allow to better mimic non-linearities between the climate and sensor and thus a truly non-linear pseudoproxy.

### 3.5.1 Ensemble of Pseudoproxies

In the following we present an ensemble of pseudoproxies. At 144 locations we compute 500 pseudoproxy records each. For this, we make slight modifications to the generalized approach. These adjustments relax our assumptions and result in larger differences between members of the ensemble than would be possible without the modifications. The locations are the grid-

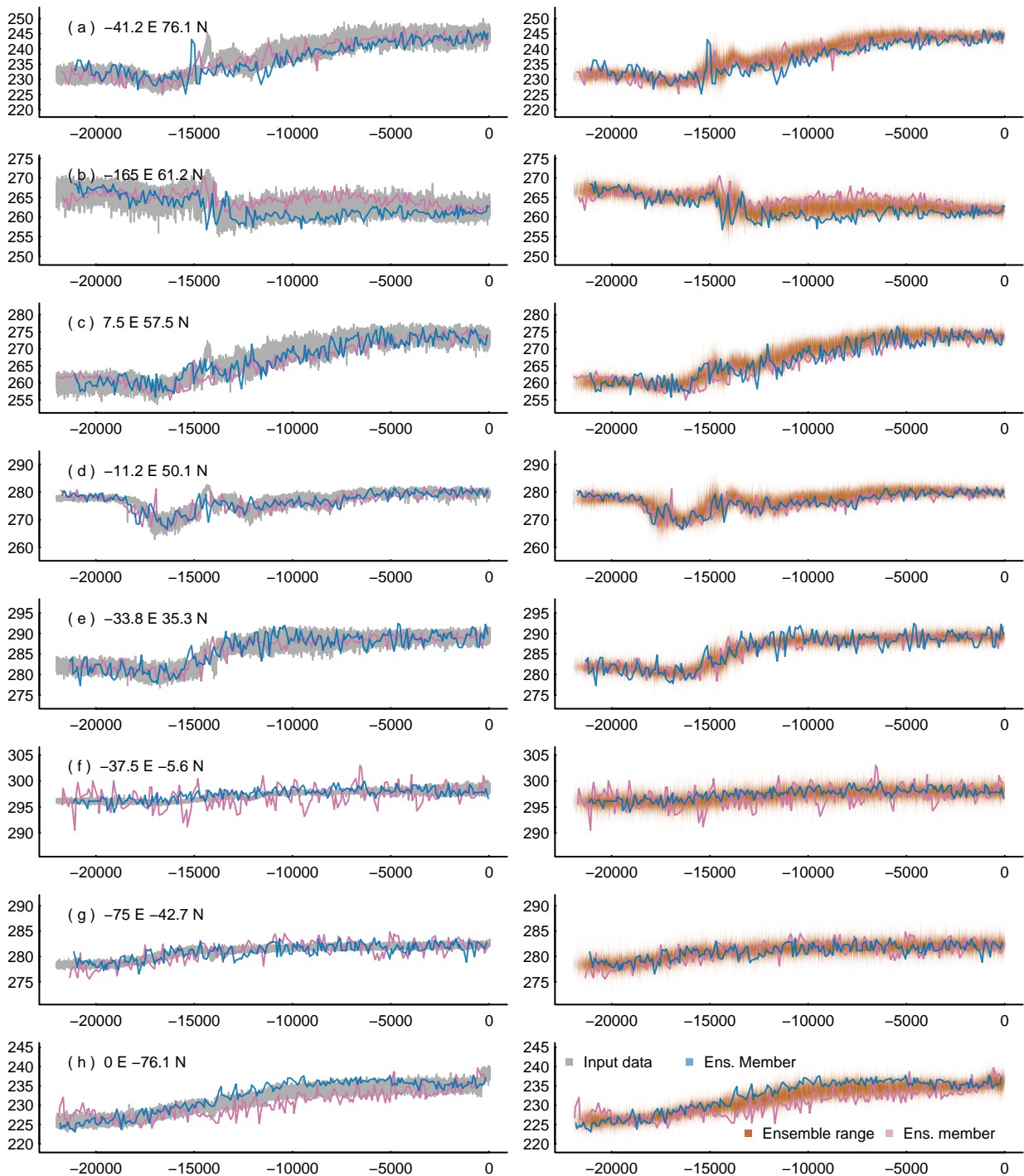

**Figure 11.** Visualising the surrogate proxy-ensemble at selected locations (Longitude and Latitude in top left corners of the left column panels): The left column shows the input data plotted as grey lines, and two random members of the ensemble as blue and purple lines. The right column plots the range of the ensemble transparently brown shaded, and blue and purple lines are the same two random members. The x-axes are years BP. The panel on the bottom right shows the Figure legend.

points, which are close to proxies either included in Shakun et al. (2012), Clark et al. (2012), or Marcott et al. (2013). Figure 10 shows the locations. Using the generalized approach provides an ensemble based on the most reduced formulation. The provided code allows users to produce ensembles for their input data of interest.

Modifications to the code are as follows: First, we use a number of parameter values sampled from either uniform distributions around the otherwise fixed value or from a list of values. Second, we consider random orientations for bias and moving standard deviations, that is we take $S$ as $S^u$ where we sample $u$ from $U = \{-1, 1\}$. We provide the script for the ensemble production as supplementary example code at https://doi.org/10.17605/OSF.IO/ZBEHX. As mentioned above, these changes relax our assumptions on the effect of changes in the background climate.

For Figure 11 we select 8 locations to represent the locally diverse representations of the climate in the TraCE-21ka simulation and how the ensemble of pseudoproxies modifies this. The Figure provides an impression of the range of the local ensembles and of two random ensemble members around the original temperature series. The diversity of the local climates in TraCE-21ka carries over to individual pseudoproxies and their ensembles. Besides this, Figure 11 mainly reflects the results of previous sections regarding how constrained our pseudoproxies are. However, we see commonly pseudoproxies and ensembles exceeding the variability of the original temperature data, not least because of our modifications to the selection of parameters and the orientation of the bias about its mean.

## 3.6 Provided Data

Tables 1 to 4 detail the provided data files. All files are in netcdf-format. These are generally gridded files on the original TraCE-21ka grid. Only the ensembles of pseudoproxies are provided at their respective individual grid-points. The data repository at https://doi.org/10.17605/OSF.IO/ZBEHX provides instructions how to access the file structures.

## 4 Conclusions and outlook

This publication presents a flexible yet simple approach for describing the error originating from climatic and non-climatic sources in proxy-records over multi-millennial time-scales including the last deglaciation. The assumptions are relatively simple but they are based on similar assumptions for process-based proxy-system forward models.

The approach can be easily extended to compute ensembles of proxies for single locations. We chose to give one set of pseudoproxies for each grid-point of the TraCE-21ka simulation and an ensemble of pseudoproxies at locations close to real proxy-locations. This simulation has a specific climatology (Liu et al., 2009) but a comparison to real proxy data may easily be achieved by only considering anomalies (as done, e.g., by Marsicek et al., 2018). The provided pseudoproxy data, and the code to compute further pseudoproxies allows the application of our pseudoproxy-approach for the evaluation of models, the comparison of models to paleo data, and the testing of reconstruction and data-assimilation methods.

We choose only one possible set of parameters in our pseudoproxy-model, but we sample around this set for the ensemble of pseudoproxies. We choose these specific parameters to provide some disturbance to the data but not to get anywhere too far

**Table 1.** List of files provided, the variables included, their description, the category (full surrogate proxy field, essence field, or ensemble), and size of the ensemble. All files have the same stem Bothe_Trace21k_Pseudo_Proxies_ and the ending _annual.nc

| Filename | Variable-Name | Variable-Description | Category | Grid size | Ensemble size |
|---|---|---|---|---|---|
| Bothe_Trace21k_Pseudo_Proxies_ | | | | | |
| noise.save | noise.save | initial environmental noise | field | 96x48 | 1 |
| bias.noise.data.save | bias.noise.data.save | data + noise + bias | field | 96x48 | 1 |
| smooth.save | smooth.save | smoothed data + noise + bias | field | 96x48 | 1 |
| meas.noise.smooth.save | meas.noise.smooth.save | smoothed data + noise + bias plus measurement noise | field | 96x48 | 1 |
| ar.smooth.save | ar.smooth.save | constantly smoothed plus AR-process | field | 96x48 | 1 |
| meas.noise.ar.smooth.save | meas.noise.ar.smooth.save | constantly smoothed plus AR plus measurement noise | field | 96x48 | 1 |
| short.smooth.save | short.smooth.save | smoothed date + noise + bias for shorter smoothing | field | 96x48 | 1 |
| meas.noise.short.smooth.save | meas.noise.short.smooth.save | smoothed data + noise + bias plus measurement noise for shorter smoothing | field | 96x48 | 1 |
| short.ar.smooth.save | short.ar.smooth.save | constantly smoothed plus AR-process for shorter smoothing | field | 96x48 | 1 |
| meas.noise.short.ar.smooth.save | meas.noise.short.ar.smooth.save | constantly smoothed plus AR plus measurement noise for shorter smoothing | field | 96x48 | 1 |
| subsampled.save | subsampled.save | seasonally subsampled data + initial noise | field | 96x48 | 1 |
| meas.noise.subsampled.save | meas.noise.subsampled.save | seasonally subsampled + noise plus measurement noise | field | 96x48 | 1 |

**Table 2.** Continued list of files provided, the variables included, their description, the category (full surrogate proxy field, essence field, or ensemble), and size of the ensemble. All files have the same stem Bothe_Trace21k_Pseudo_Proxies_ and the ending _annual.nc

| Filename | Variable-Name | Variable-Description | Category | Grid size | Ensemble size |
|---|---|---|---|---|---|
| sampled | samp.subsampled.save, samp.meas.noise.smooth.save, samp.input.save, samp.input.save.short, samp.input.save.ar, samp.input.save.ar.short, samp.noise.save, samp.noise.save.short, samp.noise.save.ar, samp.noise.save.ar.short, samp.bias.noise.data.save, samp.bias.noise.data.save.short, samp.bias.noise.data.save.ar, samp.bias.noise.data.save.ar.short, samp.ar.smooth.save, samp.smooth.save, samp.short.smooth.save, samp.short.ar.smooth.save, samp.meas.noise.short.smooth.save, samp.dates.save, samp.dates.save.short, samp.dates.save.ar,samp.dates.save.ar.short, samp.meas.noise.ar.smooth.save, samp.meas.noise.short.ar.smooth.save, samp.meas.noise.subsampled.save | sampled versions of the various variables and the dates of the samples | field | 96x48 | 1 |

**Table 3.** Continued list of files provided, the variables included, their description, the category (full surrogate proxy field, essence field, or ensemble), and size of the ensemble. All files have the same stem Bothe_Trace21k_Pseudo_Proxies_ and the ending _annual.nc

| Filename | Variable-Name | Variable-Description | Category | Grid size | Ensemble size |
|---|---|---|---|---|---|
| dating-error | samp.dates.save, samp.dates.save.short, samp.dates.save.ar,samp.dates.save.ar.short, unc.date.samp, unc.date.samp.short, unc.date.samp.ar, unc.date.samp.ar.short, unc.date.subsampled.save, unc.date.meas.noise.smooth.save, unc.date.noise.save, unc.date.bias.noise.data.save, unc.date.ar.smooth.save, unc.date.smooth.save, unc.date.short.smooth.save, unc.date.short.ar.smooth.save, unc.date.meas.noise.short.smooth.save, unc.date.meas.noise.ar.smooth.save, unc.date.meas.noise.short.ar.smooth.save, unc.samp.meas.noise.subsampled.save | date uncertain versions of the various variables and the dating uncertainties | field | 96x48 | 1 |

**Table 4.** Continued list of files provided, the variables included, their description, the category (full surrogate proxy field, essence field, or ensemble), and size of the ensemble. All files have the same stem Bothe_Trace21k_Pseudo_Proxies_ and the ending _annual.nc

| Filename | Variable-Name | Variable-Description | Category | Grid size | Ensemble size |
|---|---|---|---|---|---|
| Essence_gen.noise.env | gen.noise.env | generalized environmental noise term | essence | 96x48 | 1 |
| Essence_noise.gen.dat | noise.gen.dat | input data + generalized envrionmental noise | essence | 96x48 | 1 |
| Essence_bias.noise.gen.dat | bias.noise.gen.dat | input data + generalized noise + bias term | essence | 96x48 | 1 |
| Essence_smooth.bias.noise.gen.dat | smooth.bias.noise.gen.dat | smoothed input + noise + bias | essence | 96x48 | 1 |
| Essence_ar.smooth.bias.noise.gen.dat | ar.smooth.bias.noise.gen.dat | smoothed input + noise + bias plus AR-process | essence | 96x48 | 1 |
| Essence_uncertain-sampled | samp.ar.smooth.bias.noise.gen.dat, unc.samp.ar.smooth.bias.noise.gen.dat, meas.unc.samp.ar.smooth.bias.noise.gen.dat, unc.date.samp.gen, samp.dates.save.gen | date uncertain versions of generalized data, generalized dating uncertainty, sample dates | essence | 96x48 | 1 |
| essence_ensemble | Pseudoproxy, Dates, DateUncertainty | Surrogate proxy data, Dating, Uncertainty of Dating | ensemble | 144 | 500 |
| | Lat, Lon | Latitude, Longitude | ensemble | 144 | 1 |

away from the original state. For example, it is quite likely that we have to face larger biases in reality than represented by our choice. Users should make their own choice of parameters according to their assumptions on the various noise-contributions.

One can easily extend the chosen approach to even longer time-scales. Some modifications may be advisable considering the dating uncertainty to account for the likely sparser data further back in time, to better accommodate the increasing uncertainty, and especially to be more realistic in considering an effective dating uncertainty error for the pseudoproxy data. Similarly, we do not consider spatial correlations in the noise. Such correlations between locations are probably relevant for some noise-terms while they are probably less important for others.

We focused on the time-series approach and did not choose a probabilistic approach like, for example, Breitenbach et al. (2012) or Goswami et al. (2014). Neither, does our approach as of now explicitly link to probabilistic age-modelling approaches as described by Haslett and Parnell (2008), Blaauw and Christen (2011), or Trachsel and Telford (2017).

There are a variety of other potential approaches how to obtain simple pseudoproxies from the model data. One such example would be to consider an envelope around the model state, to select randomly a set of dates from the original data, fit a smooth through this set and then sample again around this uncertain smoothing. Similarly, Gaussian Process Models or Generalized Additive Models may be valuable means in producing pseudoproxies for paleoclimate studies over time-scales longer than the Common Era of the last 2,000 years. For example, Simpson (2018) shows the benefits of Generalized Additive Models for studies on paleoenvironmental time series.

The present approach ignores a variety of possible complications. For example, we currently do not consider hiatusses in the sensor. Furthermore, the dependency on the background climate is small. Nevertheless, we are confident that this approach is of value for the comparison of simulation data and proxy data over long periods, for testing reconstruction methods, and for evaluating different model simulations against each other.

**Table A1.** List of parameters used.

| Description | Parameter | Value | Category |
|---|---|---|---|
| Season limits for insolation bias | mon1.for.insol, mon2.for.insol | 1, 12 | all |
| Number of samples along the full record | n.samples | 200 | all |
| Scaling of initial noise amplitude | amp.noise.env | 0.5 | field, essence |
| Switch for proportionality of initial noise | switch.orient.runsd.noise.env | 0 | all |
| Model for the initial noise | model.noise.1 | c(0.3) | field, essence |
| Standard deviation of innovations for initial noise | sd.noise.1 | not used | field, essence |
| Length of window influencing initial noise | length.window.runsd | 1000 | field, essence |
| Switch for orientation of bias | switch.orient.bias.seas | 0 | all |
| Scaling of bias term | amp.bias.seas | 4 | field, essence |

## 5 Code and data availability

The TraCE-21ka simulation data is available from www.cgd.ucar.edu/ccr/TraCE and was obtained via the Earth System Grid (www.earthsystemgrid.org/project/trace.html). Our results as described in section 3.6 are available from the Open Science Framework (OSF) at https://doi.org/10.17605/OSF.IO/ZBEHX/. There, one also finds sample code for computing proxies and the script for computing the ensemble at 144 locations.

## Appendix A: Tables of parameters

Tables A1 to A4 summarise the considered parameters and noise models. They also clarify whether the parameter settings are used for a global field of surrogate proxies, a more generalized approach, an ensemble calculation, or all.

**Table A2.** Continuation of list of parameters used.

| Description | Parameter | Value | Category |
|---|---|---|---|
| Switch for smoothing variant | switch.smoothing | 3 | field |
| Secondary switch for smoothing, see code | switch.sm.2 | 1 | field |
| Scaling for climate dependence of smoothing | scale.sm | 1/10 | field |
| Mean smoothing length for longer random smoothing | rand.mean.length.smooth | 350 | field |
| Standard deviation for longer random smoothing | rand.sd.length.smooth | 75 | field |
| Model for longer alternative smoothing | model.smooth.1 | c(0.99) | field |
| Model for longer alternative climate dependent smoothing | model.clim.smooth.1 | c(0.9) | field |
| Basis long smoothing length for alternative approach | rand.length.smooth.mean.1 | 500 | field |
| Standard deviation for longer alternative smoothing approaches | sd.model.smooth.1 | 10 | field |
| Fixed longer smoothing length | fix.length.smooth | 501 | field |
| Minimum allowed longer random smoothing length | min.rand.length.smooth | 40 | field |
| AR-coefficient for added AR(1)-process | coeff.ar.smooth | 0.999 | field |
| Standard deviation for the innovations | sd.ar.smooth | 0.01 | field |
| Mean smoothing length for shorter smoothing | rand.mean.length.smooth.2 | 31 | field |
| Standard deviation for shorter random smoothing | rand.sd.length.smooth.2 | 5 | field |
| Model for shorter alternative smoothing | model.smooth.2 | c(0.7) | field |
| Model for shorter alternative climate dependent smoothing | model.clim.smooth.2 | c(0.9) | field |
| Basis short smoothing length for alternative approach | rand.length.smooth.mean.2 | 31 | field |
| Standard deviation for shorter alternative smoothing approaches | sd.model.smooth.2 | 4 | field |
| Fixed shorter smoothing length | fix.length.smooth.2 | 31 | field |
| Minimum allowed shorter random smoothing length | min.rand.length.smooth.2 | 5 | field |
| AR-coefficient for added AR(1)-process | coeff.ar.smooth.2 | 0.9 | field |
| Standard deviation for the innovations | sd.ar.smooth.2 | 0.15 | field |

**Table A3.** Continuation of list of parameters used.

| Description | Parameter | Value | Category |
| --- | --- | --- | --- |
| Number of picked samples for subsampling | n.samp.pick | 30 | field |
| Standard deviation of innovations for subsampling noise | sd.noise.pick | 0.5 | field |
| Model of subsampling noise | model.noise.pick | c() | field |
| 1.96 sigma of measurement-noise | lim.noise.meas | 1.5 | field, essence |
| Noise model for measurement noise | model.noise.meas | c() | field, essence |
| Noise model for measurement noise for subsampled record | model.seas.pick.noise.meas | c() | field |
| 1.96 sigma for measurement noise for subsampled record | lim.seas.pick.noise.meas | 1.5 | field |
| Switch for correlated effective dating error | switch.cor.date.unc | 1 | all |
| Switch for weakly correlated only | switch.weak.cor.date.unc | 1 | all |
| Switch for time dependent correlated | switch.delta.cor.date.unc | 1 | all |
| Fixed correlated dating error coefficient | cor.date.unc | 0.9 | all |
| Mean of distribution of dating uncertainty | mean.date.unc | 350 | all |
| Standard deviation of distribution of dating uncertainty | sd.date.unc | 100 | all |
| Switch for length of influence on dating uncertainty | switch.cor.length | 1 | all |
| Switch for date sampling | switch.sampling | 1 | all |
| Switch for dating uncertainty sampling | switch.sampling.unc | 1 | all |
| Model for initial noise for generalized case | model.gen.noise | c(0.7) | essence |
| Model for initial noise for generalized case | coeff.gen.ar.smooth | 0.999 | essence, ensemble |
| Standard deviation for AR process innovations, generalized case | sd.gen.ar.smooth | 0.01 | essence, ensemble |
| Smoothing length generalized case | length.filter.uniform | 501 | essence |

**Table A4.** Continuation of list of parameters used.

| Description | Parameter | Value | Category |
| --- | --- | --- | --- |
| Ensemble size | size.ensemble | 500 | ensemble |
| Amplitude of scaling of initial noise | amp.noise.env | $U(0.4, 1.5)$ | ensemble |
| Scaling of bias | amp.bias.seas | $U(3, 10)$ | ensemble |
| Standard deviation of measurement noise | lim.noise.meas | $U(0.75, 3)/1.959964$ | ensemble |
| AR-coefficient of measurement noise model | rand.model.coeff | $U(0.3, 0.8)$ | ensemble |
| AR-coefficient of initial noise model | rand.model.coeff.gen | $U(0.6, 0.8)$ | ensemble |
| Window of influence of background climate—not used | rand.width.background.sd | $U(500, 2000)$ | ensemble |
| Window of influence of background climate | rand.width.background.sd | 1000 | ensemble |
| Width of window of filter influence | length.filter.uniform | is random sample from $L = \{301, 303, 305, ..., 1001\}$ | ensemble |

*Competing interests.* The authors are not aware of any circumstances that one could see as conflicts of interest.

*Acknowledgements.* This work contributes to PalMod, German Climate Modeling Initiative: From the Last Interglacial to the Anthropocene— Modeling a Complete Glacial Cycle. It is funded through the German Federal Ministry of Education and Research's (BMBF) Research for Sustainability initiative (FONA). We acknowledge discussions with Andrew Dolman, Thom Laepple, and Nils Weitzel as influences in our approach.

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
