# Peer review of "Simple noise estimates and pseudoproxies for the last 21k years"

_Earth System Science Data, 2018_

## Referee Comment (RC1) · Anonymous Referee #1 · 18 Dec 2018

General Comments

In their manuscript, Bothe et al. provides a flexible approach to take into consideration noise in forward models of paleoclimate proxies, i.e. pseudoproxies. Although the need for these pseudo proxy models is increasing, I'm unclear as of how the present study adds to the growing body of methods available, including the recent SedProxy toolbox of Dolman and Laepple referenced by the author.

I would not recommend the manuscript for publication in this present stage. In particular, I suggest the authors make the following points clearer in their revisions: The work seems to rely on the concept of proxy system introduced by Evans et al. (2013). A proxy system is composed of an archive, a sensor, and an observation (measurement in the present manuscript). Each components can be modeled independently to ob-

tain the full proxy system model. Proxy systems model extend to age determination methods and I'm unsure as whether it was singled out in this manuscript. Ideally, in order to fully represent the uncertainty in the proxy, one would want to use a proxy system model for the time axis (e,g, radiocarbon in foraminifera shells) and y-axis (e.g., Mg/Ca in foraminiferal shell). In this particular example, the archive and sensor model would be common to the x-axis and y-axis. The observation model would need to be tailored to the particular measurement. The back and forth between age uncertainties and environmental variable uncertainties in the manuscript is confusing. I'm also unclear on how this model is fully generalizable since each type of observations made on this archive (e.g., Mg/Ca in foraminifera shells or UK37) would have specific "noise" associated with them which would need to be modeled individually. The authors keep referring to "non-climatic noise". Climatic noise is also included in the proxy records and is often impossible to disentangle from the other sources of noise discussed in the manuscript.

Specific comments:

Introduction: The concepts of proxy systems and proxy systems models need to be introduced earlier and a description of how the current work fits into these larger concepts need to be included.

Please also include a discussion on how the present approach is different from the slew of studies on proxy system models and what it adds to the table.

Page 2, line 13: A proxy system is a mathematical representation of the proxy, including the error. How is this a second way. I'm also unsure how the noise is not observation-specific? Page 2, line 30: How is a probabilistic description not a way to capture the error?

Page 4, line 3: Why choose an arbitrary point on the map? Why not a place where it would be possible to have a sedimentary record in the first place (ocean or lake)?

Page 6, line 14: I'm rather unclear about "Bias at the reconstruction level"? Aren't all sources of noise and biases important the reconstruction level? Also how can seasonality not be considered sensor uncertainty?

Page 7, line 18: The change in noise level is not obvious at all. Page 7, line 22: What three versions? Since it seems to be important, could you describe them?

Page 17: Why use the Lomb for comparison? It is know to have a bias in the high frequencies. The WWZ transform might be a better option for unevenly-spaced datasets.

Page 17, line 27: In pseudo proxy experiments, something needs to be used as ground truth would it be reanalysis data, instrumental data or in the case the TracCE-21ka output. I'm unclear as how these peaks are spurious rather than "the proxy didn't capture them."

Page 19, last three lines: I agree that a process-based model would be more useful and they are fairly simple to implement. Hence, I don't understand how the noise approach presented here is useful.

Technical corrections

I would also suggest editing the manuscript for English. For instance,

Page 2, line 3: "as base of the comparisons". Do you mean "as a basis for comparison"? Page 2, line 4: "a, eg. Temperature reconstruction and the model" is clearly missing words.

---

## Referee Comment (RC2) · Anonymous Referee #2 · 19 Dec 2018

Review of "Simple noise estimates and pseudoproxies for the last 21k years"

This study presents a generalizable approach to modeling sedimentary proxy systems and then shows how it works using the TraCE-21ka simulation. I think this is a good study that provides a flexible way to estimate various kinds of noise in proxies and that provides a nice set of pseudoproxies for potential use in a pseudo-reconstruction framework. I also think that this study can be useful for seeing how different uncertainties can affect proxy time series.

I have a number of comments, corrections, and requests for clarification below:

Abstract and elsewhere: The use of e.g. and i.e. is too frequent and would be better to just re-write with words.

There are several paragraphs throughout that are just two sentences, which is a little unusual and not totally necessary, and so would be better suited to combine with surrounding paragraphs.

Introduction: Can you better situate the present study in the context of previous approaches to generating sedimentary proxy system models/pseudoproxies? What is unique about this approach? Is it more comprehensive than previous studies? Does it innovatively use the Evans et al. 2013 framework? Is it the first to be applied to the TraCE simulation or to generate pseudoproxies over this time frame? Etc.

p.2 l.8 The words "The review" just after citing both Smerdon 2012 and Mann and Rutherford 2002 make it unclear which paper you're referring to.

p.2 l.12-17 I'm not sure this discussion of "three" different ways is quite right or at least I think I disagree with the framing of the issues here. For instance, the "proxy system model" framework of Evans et al. 2013 subsumes all of these. And so it's not as though using a proxy system model framework is a different approach from just estimating proxy error, it's that just estimating proxy error is usually considering only one of several issues that must be accounted for in the construction of pseudoproxies (i.e., only estimating the "sensor model" while potentially ignoring the "archive model" and the "observation model", using the terminology of Evans et al. 2013).

p.3 l.13-15 It's not clear to me what this sentence means. "On top of this one could use additional stages for the environment and the final reconstruction, however, we can include the associated uncertainties in any of the three stages proposed by Evans et al." The different stages have different types of noise that are particular to the specific process under consideration.

p.7 l.12-15 It's not clear to me what the bias term actually is here. You mention several different things like that it is dependent on insolation, or that it is scaled to be positive, or that it is randomized, or that it is scaled by an ad hoc constant. So what is it then? All of these at once? Only one at a time? Can you state this more clearly and/or perhaps

show in mathematical terms what you mean for the different cases?–just having the term "Bias(t)" isn't exactly clear.

Figure 6: I recommend putting the dates of the periods on this figure so it's more clear which figures correspond to what period (e.g., the deglaciation vs. the Holocene, which have very different correlation maps)

Figure 7: It would help the reader to briefly explain what the values imply. Logs of standard deviation ratios aren't necessarily intuitive. Also indicating the specific date ranges that you're using (as in my comment on Fig. 6) would be helpful.

For both Fig 6 & 7. The color map used here is usually for dry-wet data, but the figures aren't about hydroclimate at all. I would recommend using a different colormap so as to minimize any confusion.

Section 3.5 It would be helpful to write this generalized model down in mathematical terms, not just explain in words, so that readers can be sure what exactly you've done in producing Fig 8 or so that they can think about ways to adjust the generalized model.

p.21 l.11-12 This sentence isn't clear.

Section 3.5.1 Can you motivate the "modifications" you're doing here? It's not obvious to me what needs modification and why. And modifications to which approach, the full version or the generalized one? And what's the motivation for using the generalized approach vs. the full approach?

I think you also need to say more clearly what approach the ensemble of pseudoproxies is based on and why you chose one relative to the other for that dataset. Would it be possible and useful to provide pseudoproxies for both approaches?

p.21 l.14 Why are there 500 pseudoproxies at 144 locations? And are there 500 total or 500 for each of the 144 locations (and thus n = 500*144 pseudoproxies)?

Fig 10. The blue lines are hard to see here.

p.23 l.24-25 I'd recommend un-gendering this line using "their"

---

## Referee Comment (RC3) · Anonymous Referee #3 · 2 Jan 2019

SUMMARY

Bothe, Wagner and Zorita present code to produce sediment pseudo-proxy time series, i.e. a time series of a temperature variable that originates from transient climate model output and that has been modified in several stages mimicking - statistically - the processes that affect sedimentary palaeoclimate archives.

This is a timely and relevant approach and could prove useful for model data comparison in the near future with more transient paleoclimate model simulations becoming available.

GENERAL COMMENTS

- unclear aims: which properties of the data will be compared? The present formulation

only allows time-mean comparisons.

- downloading and testing the data generation is cumbersome as all parts apparently have to be manually downloaded. It would help to have a provided zip file, and a README on how to get started.

DETAILED COMMENTS

- p1 l4/following: the term "pseudoproxies" suggests that it is possible to hand the code a description of a specific sediment record (including e.g. information on the number/precision/type of dating) and all ensuing uncertainties are considered. This is not the case here, as all terms of non-climatic/insolation uncertainty considered remain statistical and non-proxy/archive specific.

-p2 l26-31: Considering dating uncertainty as purely additive white noise independent of the time axis strongly limits the suitability of the resulting time series. Autocorrelation results from the distortion of the time axis by changes in accumulation rate - which should, in a real proxy record, be captured by dating, and subsequent age modelling. Dating uncertainty represents a large component of the overall contribution to the low signal to noise ratio (c.f. Reschke, Rehfeld Laepple, Clim. Past. Discuss). The net cross-ensemble mean of the dating contribution to the final pseudoproxy uncertainty is zero in the presented formulation, as is the serial correlation of the component. Both is not appropriate. It would be beneficial to adopt (or include/prepare for) ensemble-based age models for the actual underlying proxy records; or if a simplistic solution is desired, to include the more realistic option of modeling age uncertainty by relative squeezing and stretching of the time axis.

- p 3 l30 and following: Why is only summer seasonality considered? Is this a limitation of the pseudoproxy code?

- p4 l3: why this gridpoint? The arbitrariness of this choice somewhat illustrates that it appears difficult to use this code to include knowledge on real-life proxy datasets (e.g. sedimentation rate/dating frequency/ multi-proxy configurations).

- Sec. 3: Please provide a graphical illustration of your pseudoproxy generation (e.g.

[Figure]

using a graphical model).

- p6 l26: Autocorrelation should be considered, as several of the noise components (dating, non-local climate) are expected to be autocorrelated processes. The difficulty will be in actually estimating the true autocorrelation that should be used for the noise process.

- p7 l6: Is the assumption of increasing noise variability with increasing parameter variability appropriate for all noise components? It would appear that larger climatic variations might be recorded more precisely. In the absence of information whether proxy noise is smaller or larger for higher or lower climate variability this term should be reconsidered.

- p7 l22: Can winter insolation be considered as bias?

- p8 l5 and following: How do the processes and results here compare to the approach by Dolman and Laepple (2018)?

- p10 l13: typo original

- p10 and following: The measurement error will depend on the type of sampling. To which degree is the sampling of the pseudo-proxy archive consecutive, overlapping, or spot-wise?

- p13 l12: Consider also the Bayesian Age-Depth modeling methods (e.g. OxCal, Bacon etc) which provide probability density functions of the proxy records.

- Figure 5: Please provide ensemble averages that allow to assess the spectral biases due to the proxy processes more easily.

- Figure 10: The time series are difficult to process and compare by eye. It appears in some cases there is an amplification of the apparent signal in the pseudoproxy record. Why? Where on the globe is the SD of the pseudoproxy > the SD of the climate signal?

- p28: missing section ref.

---

## Author Comment (AC1) · 1 Feb 2019

Dear referees, dear editor,

we would to like to thank you for your constructive and relevant comments on our manuscript.

In the following, we address your comments and suggestions. Our answers are in red font color while your comments are in normal font. Where feasible we will shortly outline our planned modifications to a revised manuscript for the case such revisions are invited. Where the direction of our revisions is clear, we will add tentative changes to the text.

We would like to thank you once more for your help.

[Figure]

On behalf of the authors,

Sincerely yours,

Oliver Bothe

**Response to referees**

**Referee 1:**

General Comments

In their manuscript, Bothe et al. provides a flexible approach to take into consideration noise in forward models of paleoclimate proxies, i.e. pseudoproxies. Although the need for these pseudo proxy models is increasing, I'm unclear as of how the present study adds to the growing body of methods available, including the recent SedProxy toolbox of Dolman and Laepple referenced by the author.

We appreciate the referee's concerns about the originality and usefulness of our manuscript, the chosen approach, and the data sets. We are glad that the referee at least acknowledges that current research directions require or at least benefit from the development of pseudoproxy approaches.

Traditional pseudoproxy applications over the Common Era of the last 2,000 years started from, and mainly still rely on, simple noise-based pseudoproxies. Thus, while there is the need of complex mechanistically modelled pseudoproxies, the last 20 years have shown that paleoclimatology benefits from having access to multiple pseudproxy generating algorithms.

Pseudoproxies may help in understanding proxy systems, testing reconstruction methods, evaluating simulations among another and against proxies, and in testing data assimilation techniques.

Regarding the work by Dolman and Laepple, we note that the present work developed parallel to their approach. From our point of view, our code and data complement their proxy system models from a more general point of view. Additionally, we would argue that we make differing or additional assumptions compared to Dolman and Laepple.

[Figure]

There are some major differences to the approach of Dolman and Laepple:

- The level of complexity of the different approaches

- The generalization of our approach including non-marine proxy archives.

Generally, our approach can be seen as complementary to the one of Dolman and Laepple and used as an independent source for the generation of pseudoproxies.

Having said this, we are going to argue in favor of the originality and the usefulness of our data. Originality of research may refer to hypotheses reported, methods used, and results obtained (Dirk, 1999, www.jstor.org/stable/285800). Morgan (1985, www.ncbi.nlm.nih.gov/pmc/articles/PMC1346489) reformulates originality as "independent or creative in thought or action". Research usually does not start from zero but refers to previous work. Independence and novelty, thus, are always relative and depend on the context.

In the present case of a manuscript that provides a data set, the questions could be, following Dirk (1999), how new is the method, and how new are the data (which is the result in this case). Considering the chosen approach, we would argue that we obviously rely on previous thinking on paleoproxies and indeed part of the work relies on discussions with colleagues like Dolman, Laepple, and Weitzel, which we have to acknowledge. However, to our knowledge, there is no publication presenting a simple noise-based pseudoproxy for deglacial and longer time-scales. There is definitely no publication using our specific approach, and, to our knowledge, there is no publication providing pseudoproxies or even ensembles of pseudoproxies for full simulation output fields for easy usage.

Regarding the potential usefulness of our data, there are two things to consider. First, is our data readily usable? As referee 3 notes, this is apparently not necessarily the case and we have to try to improve on this. Second, is the data by itself of value?

[Figure]

The data, as it is, can be used to test data assimilation methods as well as other reconstruction methods, to evaluate the TraCE-21ka simulation against old and new proxy data, and to test model-data comparison methods. Using our code allows to produce comparable proxies for other simulations and, e.g., additonally to compare these different simulations among another. Is our data worse or better suited than other pseudoproxies for these purposes? We provide an alternative, which we think worthwhile and usable.

I would not recommend the manuscript for publication in this present stage.

We appreciate the referee's comments and hope that our responses are sufficient to change this assessment.

In particular, I suggest the authors make the following points clearer in their revisions: The work seems to rely on the concept of proxy system introduced by Evans et al. (2013). A proxy system is composed of an archive, a sensor, and an observation (measurement in the present manuscript). Each components can be modeled independently to obtain the full proxy system model.

We are unclear what the referee refers to.

Indeed, our work relies on the concepts introduced by Evans et al. We agree, that a proxy system can be thought of as including (at least) a sensor, an archive and the observation.

If the referee implies that each has to be modelled separately, then we disagree. In our understanding, this would disqualify VSLite as proxy system model. Similarly, the approach of Thompson et al. (2011) would then not qualify either. If the referee means that it is possible to formulate models for each of the components of a proxy system in the sense of Evans et al., then we do not see their concern, as we precisely follow this approach.

Proxy systems model extend to age determination methods and I'm unsure as whether

it was singled out in this manuscript.

We are unsure what the referee refers to. Thus, our following response may miss the point.

Our aim is to provide a pseudoproxy-setup that adds a noise-based error-term for the time-uncertainty to the discrete time-series of the pseudoproxy instead of providing a tuple of uncertainties for the tuple of time and data. Our rationale is that this helps in model-evaluation, model-data comparisons, and reconstruction exercises. In a sense, we single this out as we regard this to be important. It may very well be we address a problem that is not of importance to people working on the proxies but in our understanding it is of importance for model-data comparison and evaluation of different model simulations.

Ideally, in order to fully represent the uncertainty in the proxy, one would want to use a proxy system model for the time axis (e,g, radiocarbon in foraminifera shells) and y-axis (e.g., Mg/Ca in foraminiferal shell). In this particular example, the archive and sensor model would be common to the x-axis and y-axis. The observation model would need to be tailored to the particular measurement.

If we understand the referee's point correctly, we agree on the optimal proxy system representation. However, we explicitly aim at (i) a simplified representation and (ii) a representation that results in a time-series with associated errors where the error term also accounts for date uncertainty.

The back and forth between age uncertainties and environmental variable uncertainties in the manuscript is confusing.

We are sorry for this and we will try to make our points more clearly in the manuscript by clarifying our terminology.

I'm also unclear on how this model is fully generalizable since each type of observations made on this archive (e.g., Mg/Ca in foraminifera shells or UK37) would have

specific "noise" associated with them which would need to be modeled individually.

Various sensors, various archives, and various observations, that is various proxies differ but also share common properties. In this sense we try to formulate a general model that assists in reconstruction method tests, model evaluation, and the evaluation of model-data comparisons.

The authors keep referring to "non-climatic noise". Climatic noise is also included in the proxy records and is often impossible to disentangle from the other sources of noise discussed in the manuscript.

The referee is correct and we have to clarify this.

Specific comments:

Introduction: The concepts of proxy systems and proxy systems models need to be introduced earlier and a description of how the current work fits into these larger concepts need to be included.

We introduce proxy systems and proxy system models now earlier in the introduction and also relate our approach to this setting. This requires a restructuring of the introduction, therefore we do not yet include our additions here.

Please also include a discussion on how the present approach is different from the slew of studies on proxy system models and what it adds to the table.

We discuss this more in the introduction. This requires a restructuring of the introduction, therefore we do not yet include our additions here.

Page 2, line 13: A proxy system is a mathematical representation of the proxy, including the error. How is this a second way. I'm also unsure how the noise is not observation specific?

We try to clarify our thinking as follows:

In our understanding there are various approaches to obtain pseudoproxies. These range from most comprehensive to most simplified. We can try to obtain a comprehensive representation from the environmental influences on a sensor to the measurement and implement this into a mechanistic forward model of the proxy system of interest. Such models can be more complex or they may concentrate on a core set of processes (compare the full and reduced implementations of the Vaganov-Shashkin approach to modelling tree-rings presented by, e.g., Evans et al., 2006, Tolwinski-Ward et al., 2011). That is, the first approach to obtaining pseudoproxies is process based. Other, more reduced approaches potentially ignore this mechanistic process understanding and focus on stochastic expressions of the noise that influence our inferences about past climates. Such an approach can try to formulate mathematically tractable expressions for statistical noise-terms, which represent the different processes or effects influencing the stages from the original environmental influence to our final observation [Dolman et al., in preparation]. Another way of producing pseudoproxies by focussing on stochastic noise expressions uses simple estimates of plausible errors. The different approaches can be very general or specific for certain proxy types. They can focus on one stage of the proxy system from environment to measurement or consider more or even all stages. Indeed, all these approaches fit into the conceptual descriptions of Evans et al. (2013).

Page 2, line 30: How is a probabilistic description not a way to capture the error?

The referee is correct that our formulation is unclear. We will clarify it along the following: Our interest explicitly is to include the uncertainty from the dating in a statistical noise term for a pseudoproxy time-series. Therefore, we do not consider Bayesian or Monte Carlo methods but take a simple approach to develop an error term for the dating uncertainty.

Page 4, line 3: Why choose an arbitrary point on the map? Why not a place where it

would be possible to have a sedimentary record in the first place (ocean or lake)?

The TraCE-21ka simulation has a low grid-resolution of about 3.75 times 3.75 degree. The chosen grid-point represents a good portion of the northern Iberian peninsula including the Pyrenees, where certainly a lake core would be possible. However, the choice is in so far arbitrary as we do not try to use the location of an available core. As this is an example for visualisation's sake, we do not see why this choice is critical. As we mention, the use of a land grid-point eases the readability in comparison to, e.g., the grid-point in the supplement. We consider switching to another marine grid-point.

Page 6, line 14: I'm rather unclear about "Bias at the reconstruction level"? Aren't all sources of noise and biases important the reconstruction level? Also how can seasonality not be considered sensor uncertainty?

We will clarify this. The referee is correct, in the end everything affects the reconstruction, and indeed seasonality and habitat are sensor specific factors.

Our thinking here is that there are processes for which our possibly wrong attribution is a factor at the reconstruction stage and not at the sensor stage although the processes indeed are forward factors in the evolution of the record from the date of an environmental occurrence to our reconstruction.

Page 7, line 18: The change in noise level is not obvious at all.

We will provide more pointers what we mean and refer more clearly to Figures 1a where one can see a change from a standard deviation of approximately 1 to approximately 1.5.

Page 7, line 22: What three versions? Since it seems to be important, could you describe them?

We will clarify this. We use three different amplitudes as shown in Figure 1b and also highlighted in Figure 1d.

Page 17: Why use the Lomb for comparison? It is know to have a bias in the high frequencies. The WWZ transform might be a better option for unevenly-spaced datasets.

We follow Dee et al. (2017) in using the Lomb-Scargle method. The preference of the one over the other is one of convenience. We feel confident in using the Lomb-Scargle by the given reference and reconsidering the wider literature.

We test the approach described by Mathias et al. (2004), which is related to the WWZ. Results and implications are comparable between Lomb-Scargle and this approach. We are yet undecided, which approach to include in the revised manuscript.

Page 17, line 27: In pseudo proxy experiments, something needs to be used as ground truth would it be reanalysis data, instrumental data or in the case the TracCE-21ka output. I'm unclear as how these peaks are spurious rather than "the proxy didn't capture them."

We will clarify that, in our understanding, these peaks and troughs are due to the specific forcing implementation as presented for example on http://www.cgd.ucar.edu/ccr/TraCE/. We now write: First, peaks and troughs at some location are clearly attributable to the specific implementation of the forcing in the TraCE-21ka simulation (He, 2011; see also, Liu et al., 2009). That is, these signals are not realistic but due to technical decisions in the production of the simulations.

Page 19, last three lines: I agree that a process-based model would be more useful and they are fairly simple to implement. Hence, I don't understand how the noise approach presented here is useful.

Our understanding of reconstruction methods and of simulations benefits from multitudes of approaches. The benefit of this manuscript is not least that it provides in its assets the data. It boils the processes down to noise or bias formulations. It follows pseudoproxy approaches in the Common Era, which rely on noise. As stated above, the usefulness of a data-set or an approach often may be assessed a priori, but not

always.

We hope that our clarifications in the manuscript are sufficient to convince the referee that even our simplistic noise-based approach has merit. In short, we would like to stress the complementary nature of our approach to more complex set-ups and the flexibility of the noise formulations, which allow easily adapting it to changes in our understanding and thus facilitates the performance testing of different tools of paleoclimatology.

Technical corrections

I would also suggest editing the manuscript for English.

We are going to thoroughly consider our use of English.

For instance, Page 2, line 3: "as base of the comparisons". Do you mean "as a basis for comparison"?

We are sorry for this oversight and will clarify the sentence

Page 2, line 4: "a, eg. Temperature reconstruction and the model" is clearly missing words.

We are sorry for this oversight and will repair the sentence

[Figure]

**Referee 2:**

This study presents a generalizable approach to modeling sedimentary proxy systems and then shows how it works using the TraCE-21ka simulation. I think this is a good study that provides a flexible way to estimate various kinds of noise in proxies and that provides a nice set of pseudoproxies for potential use in a pseudo-reconstruction framework. I also think that this study can be useful for seeing how different uncertainties can affect proxy time series.

We want to thank the referee for their comments and their generous evaluation of our manuscript.

I have a number of comments, corrections, and requests for clarification below:

Abstract and elsewhere: The use of e.g. and i.e. is too frequent and would be better to just re-write with words.

We correct this in the revised version.

There are several paragraphs throughout that are just two sentences, which is a little unusual and not totally necessary, and so would be better suited to combine with surrounding paragraphs.

We reorganise the manuscript accordingly.

Introduction: Can you better situate the present study in the context of previous approaches to generating sedimentary proxy system models/pseudoproxies? What is unique about this approach? Is it more comprehensive than previous studies? Does it innovatively use the Evans et al. 2013 framework? Is it the first to be applied to the TraCE simulation or to generate pseudoproxies over this time frame? Etc.

We position our manuscript, data, and approach better in the larger context. This requires a restructuring of the introduction, therefore we do not yet include our changes

here.

p.2 l.8 The words "The review" just after citing both Smerdon 2012 and Mann and Rutherford 2002 make it unclear which paper you're referring to.

We thank the referee for spotting this. We clarify this.

p.2 l.12-17 I'm not sure this discussion of "three" different ways is quite right or at least I think I disagree with the framing of the issues here. For instance, the "proxy system model" framework of Evans et al. 2013 subsumes all of these. And so it's not as though using a proxy system model framework is a different approach from just estimating proxy error, it's that just estimating proxy error is usually considering only one of several issues that must be accounted for in the construction of pseudoproxies (i.e., only estimating the "sensor model" while potentially ignoring the "archive model" and the "observation model", using the terminology of Evans et al. 2013).

We try to clarify our framing as follows:

In our understanding there are various approaches to obtain pseudoproxies. These range from most comprehensive to most simplified. We can try to obtain a comprehensive representation from the environmental influences on a sensor to the measurement and implement this into a mechanistic forward model of the proxy system of interest. Such models can be more complex or they may concentrate on a core set of processes (compare the full and reduced implementations of the Vaganov-Shashkin approach to modelling tree-rings presented by, e.g., Evans et al. 2006, and Tolwinski-Ward et al., 2011). That is, the first approach to obtaining pseudoproxies is process based. Other, more reduced approaches potentially ignore this mechanistic process understanding and focus on stochastic expressions of the noise that influence our inferences about past climates. Such an approach can try to formulate mathematically tractable expressions for statistical noise-terms, which represent the different processes or effects influencing the stages from the original environmental influence to our final observation [Dolman et al., in preparation]. Another way of producing pseudoproxies by focusing on

stochastic noise expressions uses simple estimates of plausible errors. The different approaches can be very general or specific for certain proxy types. They can focus on one stage of the proxy system from environment to measurement or consider more or even all stages. Indeed, all these approaches fit into the conceptual descriptions of Evans et al. (2013).

p.3 l.13-15 It's not clear to me what this sentence means. "On top of this one could use additional stages for the environment and the final reconstruction, however, we can include the associated uncertainties in any of the three stages proposed by Evans et al." The different stages have different types of noise that are particular to the specific process under consideration.

We are not yet clear, whether to remove or to clarify these sentences. Our thinking here is: Considering the reconstruction stage, our, e.g., calibration introduces additional uncertainty, which is not a priori captured by the stages sensor, archive, measurement. We can argue to include it in the measurement stage. We can also argue that these uncertainties are de facto uncertainties resulting from processes at the sensor stage or at the archiving stage. Similarly, our understanding is that the sensor model does not commonly account for all uncertainties of the environmental influences. That is, an additional environmental stage could provide weighted data of various environmental influences. These processes, however to some extent, can also be included in the sensor model or uncertainties can be assumed to mostly affect the measurement model.

p.7 l.12-15 It's not clear to me what the bias term actually is here. You mention several different things like that it is dependent on insolation, or that it is scaled to be positive, or that it is randomized, or that it is scaled by an ad hoc constant. So what is it then? All of these at once? Only one at a time? Can you state this more clearly and/or perhaps show in mathematical terms what you mean for the different cases?–just having the term "Bias(t)" isn't exactly clear.

In this formulation, we add one time-dependent bias-term. It is calculated dependent

on insolation, it is positive but it could be negative or the sign could be randomized, and it is scaled by an ad-hoc constant. We will clarify this and provide the equations for the bias term.

Figure 6: I recommend putting the dates of the periods on this figure so it's more clear which figures correspond to what period (e.g., the deglaciation vs. the Holocene, which have very different correlation maps)

We will clarify this.

Figure 7: It would help the reader to briefly explain what the values imply. Logs of standard deviation ratios aren't necessarily intuitive. Also indicating the specific date ranges that you're using (as in my comment on Fig. 6) would be helpful.

We will clarify and discuss this.

For both Fig 6  7. The color map used here is usually for dry-wet data, but the figures aren't about hydroclimate at all. I would recommend using a different colormap so as to minimize any confusion.

We change the color scales.

Section 3.5 It would be helpful to write this generalized model down in mathematical terms, not just explain in words, so that readers can be sure what exactly you've done in producing Fig 8 or so that they can think about ways to adjust the generalized model.

We will provide a mathematical formulation of the generalizations.

p.21 l.11-12 This sentence isn't clear.

We clarify the sentence as follows: If we repeat the analyses in Figures 6 and 7 for the generalized approach, differences are hardly to identify.

Section 3.5.1 Can you motivate the "modifications" you're doing here? It's not obvious to me what needs modification and why. And modifications to which approach, the full

version or the generalized one? And what's the motivation for using the generalized approach vs. the full approach? I think you also need to say more clearly what approach the ensemble of pseudoproxies is based on and why you chose one relative to the other for that dataset. Would it be possible and useful to provide pseudoproxies for both approaches?

We clarify all these points in the revised version.

We now write: In the following we present an ensemble of pseudproxies. At 144 locations we compute 500 pseudoproxy records each. For this, we make further slight modifications to the generalized approach. These adjustments relax our assumptions and result in larger differences between members of the ensemble than would be possible without the modifications. [...] Using the generalized approach provides an ensemble based on the most reduced formulation. [...] As mentioned above, these changes relax our assumptions on the effect of changes in the background climate.

Indeed, it would be also of value to provide pseudoproxy ensembles for both the full approach as well as the un-modified generalizations. We will consider to provide two or even three ensembles in preparing the revised manuscript.

p.21 l.14 Why are there 500 pseudoproxies at 144 locations? And are there 500 total or 500 for each of the 144 locations (and thus n = 500*144 pseudoproxies)?

We clarify this now in the text.

Indeed, there is an ensemble of 500 pseudoproxies at the 144 locations. We chose the 144 locations as the unique locations after screening proxy locations.

Fig 10. The blue lines are hard to see here.

We will clarify the Figure.

p.23 l.24-25 I'd recommend un-gendering this line using "their"

We thank the referee for spotting this extremely embarrassing mistake, and want to

apologize for it. We change this.

**Referee 3:**

SUMMARY

Bothe, Wagner and Zorita present code to produce sediment pseudo-proxy time series, i.e. a time series of a temperature variable that originates from transient climate model output and that has been modified in several stages mimicking - statistically - the processes that affect sedimentary palaeoclimate archives.

This is a timely and relevant approach and could prove useful for model data comparison in the near future with more transient paleoclimate model simulations becoming available.

We want to thank the referee for their comments and their generous evaluation of our manuscript.

GENERAL COMMENTS

- unclear aims: which properties of the data will be compared? The present formulation only allows time-mean comparisons.

We are unclear about the direction of these questions.

We will try to clarify the purpose of our pseudoproxy data.

The provided data and the code allow for the generation of pseudoproxy records for any simulation. These can be continuous or temporally sparse and regularly or irregularly sampled. The provided data particularly allows the comparison of different models against another over time, as well as testing methods for the comparison of simulation and proxy data over time. The irregular sampling however hampers the comparison of time-slices.
As said, we are unclear about the direction of this question/comment.

- downloading and testing the data generation is cumbersome as all parts apparently have to be manually downloaded. It would help to have a provided zip file, and a README on how to get started.

We will provide one or more zip-files collecting the different types of data provided. A short documentation will clarify how to best approach data and code-examples.

DETAILED COMMENTS

- p1 l4/following: the term "pseudoproxies" suggests that it is possible to hand the code a description of a specific sediment record (including e.g. information on the number/precision/type of dating) and all ensuing uncertainties are considered. This is not the case here, as all terms of non-climatic/insolation uncertainty considered remain statistical and non-proxy/archive specific.

We understand and respect the referee's point but after reconsidering the term and its use in the literature, we disagree. The term pseudoproxy does not refer to a surrogate for a specific record. In the literature, pseudoproxy, surrogate proxy, and pseudoproxy experiments are phrases, which refer to modifications of observational data, reanalysis data, or simulation output. The applications of such modifications are not limited to either real world proxy records or real world proxy types. Such modifications in the broad sense of pseudoproxies are simply stand-ins for real world data in research enterprises of interest.

The revised manuscript explains our view of the term pseudoproxy. We add in the introduction the following: We clarify here our use of the term pseudoproxy. We follow the literature since Mann and Rutherford (2002). That is, a pseudoproxy simply represents a modification of observational data, reanalysis data, or simulation output. It replaces real world proxies in a certain application. The term may but does not nec-
essarily refer to stand-ins for specific proxy records or particular proxy types. That is, the term pseudoproxy does not by itself imply that the modifications of the input data represent validly the uncertainties or characteristics of real world data. This view of the term pseudoproxy is in line with the past literature (compare, for example, Mann and Rutherford, 2002; Osborn and Briffa, 2004; von Storch et al., 2004; Jones et al., 2009; Graham and Wahl, 2011; Thompson et al., 2011; Lehner et al., 2012; Smerdon, 2012; Hind et al., 2012; Annan and Hargreaves, 2013; Kurahashi-Nakamura et al., 2014; Steiger and Hakim, 2016). These modifications may be simply by adding noise to the input data or may invoke more complex forward approaches (for example mechanistic Proxy System Models, Evans et al., 2013, see below).

-p2 l26-31: Considering dating uncertainty as purely additive white noise independent of the time axis strongly limits the suitability of the resulting time series. Autocorrelation results from the distortion of the time axis by changes in accumulation rate - which should, in a real proxy record, be captured by dating, and subsequent age modelling. Dating uncertainty represents a large component of the overall contribution to the low signal to noise ratio (c.f. Reschke, Rehfeld Laepple, Clim. Past. Discuss). The net cross-ensemble mean of the dating contribution to the final pseudoproxy uncertainty is zero in the presented formulation, as is the serial correlation of the component. Both is not appropriate. It would be beneficial to adopt (or include/prepare for) ensemble-based age models for the actual underlying proxy records; or if a simplistic solution is desired, to include the more realistic option of modeling age uncertainty by relative squeezing and stretching of the time axis.

We appreciate the referee's concerns. There are various points here. We disagree on the suitability of the resulting time-series. We are still confident that the time-series including the error terms are suitable for wide range of applications in, e.g., comparing different model simulations, model-data evaluation, and testing of reconstruction methods. The dating uncertainty in our set-up has two parts, the sampling of the dates and the sampling of the dating uncertainty on the one hand, and the associated dating error

modelling on the other hand.

Indeed, not all our implementations of the dating error showed serial correlation so far, however, most and particularly the main approach led to a small amount of serial correlation by making the dating uncertainty error dependent on previous errors and "measurements".

We will modify our code to allow for larger uncertainty and correlation in the final products. The specific implementation is still work in progress, but modifications are possible for the sampling of the dates, the date uncertainties, or the correlation between modelled errors. However, if tests of this new approach fail to show an added value we are going to fall back to the current version for our data but the modifications will be kept in the code. As mentioned above, we still regard the full set of pseudoproxies of immediate value for a number of tasks.

Thus, indeed, we choose to concentrate on a simplistic approach.

- p 3 l30 and following: Why is only summer seasonality considered? Is this a limitation of the pseudoproxy code?

We concentrate on the modern boreal summer season for convenience sake and not least since many authors attribute the sensitivity of their proxies to the summer season. We could have similarly used annual data or any other seasonal definition. The pseudoproxy code takes a time-series of any seasonality of interest. At this stage, it does not read the full annual matrix. We will consider within the preparation of revisions to formulate the code more flexibly.

- p4 l3: why this gridpoint? The arbitrariness of this choice somewhat illustrates that it appears difficult to use this code to include knowledge on real-life proxy datasets (e.g. sedimentation rate/dating frequency/ multi-proxy configurations).

As the referee already notes, this decision was generally arbitrary. It was made in relation to comparison with various proxies around the Iberian Peninsula and more

generally Europe in another context. As we describe, the visualisation for another proxy is provided in the Supplementary materials. As we state above, the aim here is not to test real world proxies but to provide data for testing methods and code to evaluate models and data. We consider changing the grid point. However, we want to stress that our approach does not aim at mimicking real proxies but at providing pseudoproxies also for locations where we do not have real world knowledge.

- Sec. 3: Please provide a graphical illustration of your pseudoproxy generation (e.g. using a graphical model).

We will provide a visualisation of the procedure but plan to put it only in a supplement or appendix.

- p6 l26: Autocorrelation should be considered, as several of the noise components (dating, non-local climate) are expected to be autocorrelated processes. The difficulty will be in actually estimating the true autocorrelation that should be used for the noise process.

As we note, the code provided allows for generating autocorrelated noise. We now change various components to correlated models.

- p7 l6: Is the assumption of increasing noise variability with increasing parameter variability appropriate for all noise components? It would appear that larger climatic variations might be recorded more precisely. In the absence of information whether proxy noise is smaller or larger for higher or lower climate variability this term should be reconsidered.

We will introduce a random switch.

While larger variations may be recorded more precisely one may also assume that larger, e.g., temperature variations are associated with larger variations in other environmental components that may result in larger errors.

- p7 l22: Can winter insolation be considered as bias?

If we understand the referees question correctly, then, yes, there is no reason why one should not consider winter insolation.

For now it is not implemented directly but we consider to change this in the revisions.

- p8 l5 and following: How do the processes and results here compare to the approach by Dolman and Laepple (2018)?

Our assumptions simplify the more complex approach of Dolman and Laepple. Our randomization could be seen as more realistic while the lack of process based assumptions make our approach less realistic.

Generally our results are not as smooth as the results of Dolman and Laepple considering the bioturbation while the assumptions on the subsampling appear to be comparable in our reading.

- p10 l13: typo original

We want to thank the referee for spotting this. We change this in the revised version.

- p10 and following: The measurement error will depend on the type of sampling. To which degree is the sampling of the pseudo-proxy archive consecutive, overlapping, or spot-wise?

In this implementation, we only consider a conceptual measurement error and sample the time-series at certain years in the time-series as stated in the text. We will clarify this in the text.

- p13 l12: Consider also the Bayesian Age-Depth modeling methods (e.g. OxCal, Bacon etc) which provide probability density functions of the proxy records.

We aim to include a brief note on age-depth modelling methods.

- Figure 5: Please provide ensemble averages that allow to assess the spectral biases due to the proxy processes more easily.

We, at this point, do not use ensembles but only the single estimate. We will consider clarifying this Figure and its description in line with the comment of referee 1 on the Lomb-Scargle periodogram.

- Figure 10: The time series are difficult to process and compare by eye. It appears in some cases there is an amplification of the apparent signal in the pseudoproxy record. Why? Where on the globe is the SD of the pseudoproxy > the SD of the climate signal?

This is mainly due to the size of the considered bias and the amplitude of noise processes. We will provide a visualisation equivalent to Figure 7 to answer the latter question.

- p28: missing section ref.

We want to thank the referee for spotting this. We change this in the revised version.

---

## Author Response (AR1)

**Response to referees ESSD-2018-137**

*28 March 2019*

Dear referees, dear editor,

we would to like to thank you for your constructive and relevant comments on our manuscript.

In the following, we reiterate or modify our initial response and indicate the changes to the manuscript. Our answers are in red font color while referees' comments are in normal font. Where feasible, we indicate changes to the manuscript in blue.

We would like to thank you once more for your help.

On behalf of the authors,

Sincerely yours,

Oliver Bothe

**List Of Changes:**

**Introduction**

Added discussion on pseudoproxies

Extended introduction of proxy system models

Extended on differences between our approach and previous works

**Data**

We changed the presented grid-point in the main manuscript and the supplementary file.

We now consider annual data instead of summer data.

**Considerations and Results**

We added a flow chart of our procedure.

We changed our code and provide more options for the various steps of our procedure.

We changed some of the noise models to allow for (higher) autocorrelations.

Particularly we clarified the questions by the referees.

We changed the spectral power estimation to a wavelet procedure.

We now show standard deviation ratios instead of their logarithm.

We added a mathematical expression for the generalized approach.

We changed the visualisation of the ensemble data.

**Conclusions**

We slightly extended our outlook.

**Generally**

We updated the tables and the supplements.

**Response to referees**

**Referee 1:**

**General Comments**

In their manuscript, Bothe et al. provides a flexible approach to take into consideration noise in forward models of paleo-climate proxies, i.e. pseudoproxies. Although the need for these pseudo proxy models is increasing, I'm unclear as of how the present study adds to the growing body of methods available, including the recent SedProxy toolbox of Dolman and Laepple referenced by the author.

We appreciate the referee's concerns about the originality and usefulness of our manuscript, the chosen approach, and the data sets. We appreciate that the referee acknowledges that current research directions require and benefit from the development of pseudoproxy approaches.

Traditional pseudoproxy applications over the Common Era of the last 2,000 years started from, and mainly still rely on, simple noise-based pseudoproxies. Thus, while there is the need of complex mechanistically modelled pseudoproxies, the last 20 years have shown that paleoclimatology benefits from having access to multiple pseudproxy generating algorithms.

Pseudoproxies may help in understanding proxy systems, testing reconstruction methods, evaluating and comparing simulations among another and against proxies, and in testing data assimilation techniques.

Regarding the work by Dolman and Laepple (2018), we note that the present work developed parallel to their approach. From our point of view, our code and data complement their proxy system models from a more general point of view. Additionally, we would argue that we make differing or additional assumptions compared to Dolman and Laepple (2018).

There are some major differences to the approach of Dolman and Laepple:

- The level of complexity of the different approaches

- The generalization of our approach including non-marine proxy archives.

Generally, our approach can be seen as complementary to the one of Dolman and Laepple and used as an independent source for the generation of pseudoproxies.

We are going to argue in favor of the originality and the usefulness of our approach. Originality of research may refer to hypotheses reported, methods used, and results obtained (Dirk, 1999, www.jstor.org/stable/285800). Morgan (1985, www.ncbi.nlm.nih.gov/pmc/articles/PMC1346489) reformulates originality as "independent or creative in thought or action". Research usually does not start from zero but refers to previous work. Independence and novelty, thus, are always relative and depend on the context.

In the present case of a manuscript that provides a data set, the questions could be, following Dirk (1999), how new is the method, and how new are the data (which is the result in this case). Considering the chosen approach, we would argue that we obviously rely on previous thinking on paleoproxies and indeed part of the work relies on discussions with colleagues like Dolman, Laepple, and Weitzel, which we have to acknowledge. However, to our knowledge, there is no publication presenting a simple noise-based pseudoproxy for deglacial and longer time-scales. There is definitely no publication using our specific approach, and, to our knowledge, there is no publication providing pseudoproxies or even ensembles of pseudoproxies for full simulation output fields for easy usage.

Regarding the potential usefulness of our data, there are two things to consider. First, is our data readily usable? As referee 3 notes, this is apparently not necessarily the case and we have to try to improve on this. Second, is the data by itself of value? The data, as it is, can be used to test data assimilation methods as well as other reconstruction methods, to evaluate the TraCE-21ka simulation against old and new proxy data, and to test model-data comparison methods. Using our code allows to produce comparable proxies for other simulations and, e.g., additonally to compare these different simulations among another. Is our data worse or better suited than other pseudoproxies for these purposes? We provide an alternative, which we think worthwhile and usable.

I would not recommend the manuscript for publication in this present stage.

We appreciate the referee's comments and hope that our responses are sufficient to change this assessment.

In particular, I suggest the authors make the following points clearer in their revisions: The work seems to rely on the concept of proxy system introduced by Evans et al. (2013). A proxy system is composed of an archive, a sensor, and an

observation (measurement in the present manuscript). Each components can be modeled independently to obtain the full proxy system model.

Indeed, our work relies on the concepts introduced by Evans et al. We agree, that a proxy system can be thought of as including (at least) a sensor, an archive and the observation.

If the referee implies that each has to be modelled separately, then we disagree. In our understanding, this would disqualify VSLite as proxy system model. Similarly, the approach of Thompson et al. (2011) would then not qualify either. If the referee means that it is possible to formulate models for each of the components of a proxy system in the sense of Evans et al., then we do not see their concern, as we precisely follow this approach.

In response to referee 3, we add a flowchart of our procedures, which hopefully clarifies the relation of our procedure to the conceptual ideas described by Evans et al. Furthermore, we more thoroughly introduce the concepts of a proxy system, of proxy system models, and of pseudoproxies.

Proxy systems model extend to age determination methods and I'm unsure as whether it was singled out in this manuscript.

We are unsure what the referee refers to. Thus, our following response may miss the point.

Our aim is to provide a pseudoproxy-setup that adds a noise-based error-term for the time-uncertainty to the discrete time-series of the pseudoproxy instead of providing a tuple of uncertainties for the tuple of time and data. Our rationale is that this helps in model-evaluation, model-data comparisons, and reconstruction exercises. In a sense, we single this out as we regard this to be important. In our understanding this is of importance for model-data comparison and evaluation of different model simulations.

Ideally, in order to fully represent the uncertainty in the proxy, one would want to use a proxy system model for the time axis (e,g, radiocarbon in foraminifera shells) and y-axis (e.g., Mg/Ca in foraminiferal shell). In this particular example, the archive and sensor model would be common to the x-axis and y-axis. The observation model would need to be tailored to the particular measurement.

We agree on the optimal proxy system representation. However, we explicitly aim at (i) a simplified representation and (ii) a representation that results in a time-series with associated errors where the error term also accounts for date uncertainty.

The code does not explicitly model the time axis. However, the sampling of dates and the assignment of uncertainties should allow in principle to apply an age modelling approach comparable to PRYSM 2.0 in Dee et al. (2018).

The back and forth between age uncertainties and environmental variable uncertainties in the manuscript is confusing.

We are sorry for this and we will try to make our points more clearly in the revised version of the manuscript by clarifying our terminology.

We tried to be as explicit as possible in distinguishing between the age uncertainties, the environmental uncertainties, and what we call the "effective dating uncertainty error.

I'm also unclear on how this model is fully generalizable since each type of observations made on this archive (e.g., Mg/Ca in foraminifera shells or UK37) would have specific "noise" associated with them which would need to be modeled individually.

Various sensors, various archives, and various observations, that is various proxies differ but also share common properties. In this sense we try to formulate a general model that assists in reconstruction method tests, model evaluation, and the evaluation of model-data comparisons.

We our confident, that the code for the various components is flexible enough to allow individual researchers to adapt the noise levels according to their understanding.

The authors keep referring to "non-climatic noise". Climatic noise is also included in the proxy records and is often impossible to disentangle from the other sources of noise discussed in the manuscript.

The referee is correct and we have to clarify this.

We tried to correct all instances of non-climatic noise.

**Specific comments:**

Introduction:

The concepts of proxy systems and proxy systems models need to be introduced earlier and a description of how the current work fits into these larger concepts need to be included.

We introduce proxy systems and proxy system models now earlier in the introduction and also relate our approach to this setting.

Please also include a discussion on how the present approach is different from the slew of studies on proxy system models and what it adds to the table.

We shortly discuss this more in the introduction.

Page 2, line 13: A proxy system is a mathematical representation of the proxy, including the error. How is this a second way. I'm also unsure how the noise is not observation specific?

We try to clarify our thinking as follows:

In our understanding there are various approaches to obtain pseudoproxies. These range from comprehensive to simplified. We can try to obtain a comprehensive representation from the environmental influences on a sensor to the measurement and implement this into a mechanistic forward model of the proxy system of interest. Such models can be more complex or they may concentrate on a core set of processes (compare the full and reduced implementations of the Vaganov-Shashkin approach to modelling tree-rings presented by, e.g., Evans et al., 2006, Tolwinski-Ward et al., 2011). That is, the first approach to obtaining pseudoproxies is process based. Other, more reduced approaches potentially ignore this mechanistic process understanding and focus on stochastic expressions of the noise that influence our inferences about past climates. Such an approach can try to formulate mathematically tractable expressions for statistical noise-terms, which represent the different processes or effects influencing the stages from the original environmental influence to our final observation and reconstruction [Dolman et al., in preparation]. Another way of producing pseudoproxies by focussing on stochastic noise expressions uses simple estimates of plausible errors. The different approaches can be very general or specific for certain proxy types. They can focus on one stage of the proxy system from environment to measurement or consider multiple stages. Indeed, all these approaches fit into the conceptual descriptions of Evans et al. (2013).

The manuscript now includes a paragraph comparable to the above initial response.

Page 2, line 30: How is a probabilistic description not a way to capture the error?

The referee is correct that our formulation is unclear. We will clarify it along the following: Our interest explicitly is to include the uncertainty from the dating in a statistical noise term for a pseudoproxy time-series. Therefore, we do not consider Bayesian or Monte Carlo methods but take a simple approach to develop an error term for the dating uncertainty.

The manuscript now includes sentences comparable to the above initial response.

Page 4, line 3: Why choose an arbitrary point on the map? Why not a place where it would be possible to have a sedimentary record in the first place (ocean or lake)?

The TraCE-21ka simulation has a low grid-resolution of about 3.75 times 3.75 degree. The chosen grid-point represents a good portion of the northern Iberian peninsula including the Pyrenees, where a lake core could be located. However, the choice is in so far arbitrary as we do not try to use the location of an available core. As this is an example for visualisation's sake, we do not see why this choice is critical. As we mentioned, the use of a land grid-point eases the readability in comparison to, e.g., the grid-point in the supplement.

We change the presented data to the grid-point at 150E, 38.97N.

Page 6, line 14: I'm rather unclear about "Bias at the reconstruction level"? Aren't all sources of noise and biases important the reconstruction level? Also how can seasonality not be considered sensor uncertainty?

We will clarify this. The referee is correct, as all sources of error affect the reconstruction, and indeed seasonality and habitat are sensor specific factors.

Our rationale here is that there are processes for which our possibly wrong attribution is a factor at the reconstruction stage and not at the sensor stage although the processes indeed are forward factors in the evolution of the record from the date of an environmental occurrence to our reconstruction.

Page 7, line 18: The change in noise level is not obvious at all.

Since we changed the grid-point, we also hope the change in the noise-level becomes more obvious. We try to provide more pointers in the text to Figures 1a.

Page 7, line 22: What three versions? Since it seems to be important, could you describe them?

We use three different amplitudes as shown in Figure 1b and also highlighted in Figure 1d.

We change "versions" to "potential amplitudes" to clarify this point.

Page 17: Why use the Lomb for comparison? It is know to have a bias in the high frequencies. The WWZ transform might be a better option for unevenly-spaced datasets.

We followed Dee et al. (2017) in using the Lomb-Scargle method. The preference of the one over the other is a matter of subjective choice in our opinion. We feel confident in using the Lomb-Scargle by the given reference and reconsidering the wider literature.

We now use the approach described by Mathias et al. (2004), which is related to the WWZ. Results and implications are comparable between Lomb-Scargle and this approach.

Page 17, line 27: In pseudo proxy experiments, something needs to be used as ground truth would it be reanalysis data, instrumental data or in the case the TracCE-21ka output. I'm unclear as how these peaks are spurious rather than "the proxy didn't capture them."

We will clarify that, in our understanding, these peaks and troughs are due to the specific forcing implementation as presented for example on http://www.cgd.ucar.edu/ccr/TraCE/.

The manuscript now includes a comment comparable to the following: First, peaks and troughs at some location are clearly attributable to the specific implementation of the forcing in the TraCE-21ka simulation (He, 2011; see also, Liu et al., 2009). That is, these signals are not realistic but due to technical decisions in the production of the simulations.

Page 19, last three lines: I agree that a process-based model would be more useful and they are fairly simple to implement. Hence, I don't understand how the noise approach presented here is useful.

Our understanding of reconstruction methods and of simulations benefits from multitudes of approaches. The benefit of this manuscript is that it provides in its assets the data. It boils the processes down to noise or bias formulations. It follows pseudoproxy approaches used for the period of the Common Era, which rely on noise. As stated above, the usefulness of a data-set or an approach often may be assessed a priori, but not always.

We hope that our clarifications in the manuscript are sufficient to convince the referee that our simplistic noise-based approach has merit. In short, we would like to stress the complementary nature of our approach to more complex set-ups and the flexibility of the noise formulations, which allow easily adapting it to changes in our understanding and thus facilitates the performance testing of different tools of paleoclimatology.

**Technical corrections**

I would also suggest editing the manuscript for English.

We tried to improve on our use of English.

For instance, Page 2, line 3: "as base of the comparisons". Do you mean "as a basis for comparison"?

We are sorry for this oversight.

We clarified the sentence.

Page 2, line 4: "a, eg. Temperature reconstruction and the model" is clearly missing words.

We are sorry for this oversight.

We modified the sentence.

**Referee 2:**

This study presents a generalizable approach to modeling sedimentary proxy systems and then shows how it works using the TraCE-21ka simulation. I think this is a good study that provides a flexible way to estimate various kinds of noise in proxies and that provides a nice set of pseudoproxies for potential use in a pseudo-reconstruction framework. I also think that this study can be useful for seeing how different uncertainties can affect proxy time series.

We would like to thank the referee for their comments and their generous evaluation of our manuscript.

I have a number of comments, corrections, and requests for clarification below:

Abstract and elsewhere: The use of e.g. and i.e. is too frequent and would be better to just re-write with words.

We hope we corrected this sufficiently in the revised version.

There are several paragraphs throughout that are just two sentences, which is a little unusual and not totally necessary, and so would be better suited to combine with surrounding paragraphs.

We reorganised the manuscript accordingly.

Introduction: Can you better situate the present study in the context of previous approaches to generating sedimentary proxy system models/pseudoproxies? What is unique about this approach? Is it more comprehensive than previous studies? Does it innovatively use the Evans et al. 2013 framework? Is it the first to be applied to the TraCE simulation or to generate pseudoproxies over this time frame? Etc.

We position our manuscript, data, and approach better in the larger context.

The introduction now tries to clarify the difference between our approach and previous applications, our contribution to this topic, how it relates to the framework of Evans et al. (2013), and how we use the TraCE simulation differently to other studies.

p.2 l.8 The words "The review" just after citing both Smerdon 2012 and Mann and Rutherford 2002 make it unclear which paper you're referring to.

We thank the referee for spotting this.

We clarify this.

p.2 l.12-17 I'm not sure this discussion of "three" different ways is quite right or at least I think I disagree with the framing of the issues here. For instance, the "proxy system model" framework of Evans et al. 2013 subsumes all of these. And so it's not as though using a proxy system model framework is a different approach from just estimating proxy error, it's that just estimating proxy error is usually considering only one of several issues that must be accounted for in the construction of pseudoproxies (i.e., only estimating the "sensor model" while potentially ignoring the "archive model" and the "observation model", using the terminology of Evans et al. 2013).

We try to clarify our framing as follows:

In our understanding there are various approaches to obtain pseudoproxies. These range from most comprehensive to most simplified. We can try to obtain a comprehensive representation from the environmental influences on a sensor to the measurement and implement this into a mechanistic forward model of the proxy system of interest. Such models can be more complex or they may concentrate on a core set of processes (compare the full and reduced implementations of the Vaganov-Shashkin approach to modelling tree-rings presented by, e.g., Evans et al. 2006, and Tolwinski-Ward et al., 2011). That is, the first approach to obtaining pseudoproxies is process based. Other, more reduced approaches potentially ignore this mechanistic process understanding and focus on stochastic expressions of the noise that influence our inferences about past climates. Such an approach can try to formulate mathematically tractable expressions for statistical noise-terms, which represent the different processes or effects influencing the stages from the original environmental influence to our final observation [Dolman et al., in preparation]. Another way of producing pseudoproxies by focusing on stochastic noise expressions uses simple estimates of plausible errors. The different approaches can be very general or specific for certain proxy types. They can focus on one stage of the proxy system from environment to measurement or consider more or even all stages. Indeed, all these approaches fit into the conceptual descriptions of Evans et al. (2013).

The manuscript now includes a justification comparable to the paragraph above.

p.3 l.13-15 It's not clear to me what this sentence means. "On top of this one could use additional stages for the environment and the final reconstruction, however, we can include the associated uncertainties in any of the three stages proposed by Evans et al." The different stages have different types of noise that are particular to the specific process under consideration.

Our thinking here is: Considering the reconstruction stage, our, e.g., calibration introduces additional uncertainty, which is not a priori captured by the stages sensor, archive, measurement. We can argue to include it in the measurement stage. We can also argue that these uncertainties are de facto uncertainties resulting from processes at the sensor stage or at the archiving stage. Similarly, our understanding is that the sensor model does not commonly account for all uncertainties of the environmental influences. That is, an additional environmental stage could provide weighted data of various environmental influences. These processes, however to some extent, can also be included in the sensor model or uncertainties can be assumed to mostly affect the measurement model.

The introduction now tries to clarify these points.

p.7 l.12-15 It's not clear to me what the bias term actually is here. You mention several different things like that it is dependent on insolation, or that it is scaled to be positive, or that it is randomized, or that it is scaled by an ad hoc constant. So what is it then? All of these at once? Only one at a time? Can you state this more clearly and/or perhaps show in mathematical terms what you mean for the different cases?–just having the term "Bias(t)" isn't exactly clear.

In the initial formulation, we add one bias-term, which varies with time. It is calculated dependent on insolation, it is positive but it could be negative or the sign could be randomized, and it is scaled by an ad-hoc constant. We will clarify this and provide the equations for the bias term.

Figure 6: I recommend putting the dates of the periods on this figure so it's more clear which figures correspond to what period (e.g., the deglaciation vs. the Holocene, which have very different correlation maps)

We add titles to clarify the Figure.

Figure 7: It would help the reader to briefly explain what the values imply. Logs of standard deviation ratios aren't necessarily intuitive. Also indicating the specific date ranges that you're using (as in my comment on Fig. 6) would be helpful.

We now visualize standard deviation ratios in the Figure. We also added titles.

For both Fig 6 7. The color map used here is usually for dry-wet data, but the figures aren't about hydroclimate at all. I would recommend using a different colormap so as to minimize any confusion.

We change the color scales.

Section 3.5 It would be helpful to write this generalized model down in mathematical terms, not just explain in words, so that readers can be sure what exactly you've done in producing Fig 8 or so that they can think about ways to adjust the generalized model.

We provide a mathematical formulation of the generalizations.

p.21 l.11-12 This sentence isn't clear.

We clarify the sentence as follows: If we repeat the analyses in Figures 6 and 7 for the generalized approach, differences are hardly to identify.

Section 3.5.1 Can you motivate the "modifications" you're doing here? It's not obvious to me what needs modification and why. And modifications to which approach, the full version or the generalized one? And what's the motivation for using the generalized approach vs. the full approach? I think you also need to say more clearly what approach the ensemble of pseudoproxies is based on and why you chose one relative to the other for that dataset. Would it be possible and useful to provide pseudoproxies for both approaches?

We clarify all these points in the revised version.

We now write something comparable to: In the following we present an ensemble of pseudproxies. At 144 locations we compute 500 pseudoproxy records each. For this, we make further slight modifications to the generalized approach. These adjustments relax our assumptions and result in larger differences between members of the ensemble than would be possible without the modifications. [...] Using the generalized approach provides an ensemble based on the most reduced formulation. [...] As mentioned above, these changes relax our assumptions on the effect of changes in the background climate.

Indeed, it would be also of value to provide pseudoproxy ensembles for both the full approach as well as the un-modified generalizations.

We decide not to provide an ensemble for the full approach.

p.21 l.14 Why are there 500 pseudoproxies at 144 locations? And are there 500 total or 500 for each of the 144 locations (and thus n = 500*144 pseudoproxies)?

We clarify this now in the text.

Indeed, there is an ensemble of 500 pseudoproxies at the 144 locations. We chose the 144 locations as the unique locations after screening proxy locations from a number of publications.

Fig 10. The blue lines are hard to see here.

We changed the layout of the Figure and hope it is clearer now..

p.23 l.24-25 I'd recommend un-gendering this line using "their"

We thank the referee for spotting this extremely embarrassing mistake, and want to apologize for it.

We change this.

**Referee 3:**

**SUMMARY**

Bothe, Wagner and Zorita present code to produce sediment pseudo-proxy time series, i.e. a time series of a temperature variable that originates from transient climate model output and that has been modified in several stages mimicking - statistically - the processes that affect sedimentary palaeoclimate archives. This is a timely and relevant approach and could prove useful for model data comparison in the near future with more transient paleoclimate model simulations becoming available.

We want to thank the referee for their comments and their generous evaluation of our manuscript.

**GENERAL COMMENTS**

- unclear aims: which properties of the data will be compared? The present formulation only allows time-mean comparisons.

We are unclear about the direction of these questions.

We will try to clarify the purpose of our pseudoproxy data.

The provided data and the code allow for the generation of pseudoproxy records for any simulation. These can be continuous or temporally sparse and regularly or irregularly sampled. The provided data particularly allows the comparison of different models against another over time, as well as testing methods for the comparison of simulation and proxy data over time. The irregular sampling however hampers the comparison of time-slices. As said, we are unclear about the direction of this question/comment.

We hope the revised manuscript clarifies our intentions for the data.

- downloading and testing the data generation is cumbersome as all parts apparently have to be manually downloaded. It would help to have a provided zip file, and a README on how to get started.

The repository allows to download folders and storage-items as zip-files. However, we provide zip-files collecting the different types of data. Note that these zip-files can become huge. A short documentation clarifies how to best approach data and code-examples. This is additionally included in the zip-files. We hope the data is now more easily accessible.

**DETAILED COMMENTS**

- p1 l4/following: the term "pseudoproxies" suggests that it is possible to hand the code a description of a specific sediment record (including e.g. information on the number/precision/type of dating) and all ensuing uncertainties are considered. This is not the case here, as all terms of non-climatic/insolation uncertainty considered remain statistical and non-proxy/archive specific.

We understand and respect the referee's point but after reconsidering the term and its use in the literature, we try to explain our view in using the term. The term pseudoproxy does not refer to a surrogate for a specific record. In the literature, pseudoproxy, surrogate proxy, and pseudoproxy experiments are phrases, which refer to modifications of observational data, reanalysis data, or simulation output. The applications of such modifications are not limited to either real world proxy records or real world proxy types. Such modifications in the broad sense of pseudoproxies are simply stand-ins for real world data in research enterprises of interest.

The revised manuscript explains our view of the term pseudoproxy. We add in the introduction a comment similar to the following: We clarify here our use of the term pseudoproxy. We follow the literature since Mann and Rutherford (2002). That is, a pseudoproxy simply represents a modification of observational data, reanalysis data, or simulation output. It replaces real world proxies in a certain application. The term may but does not necessarily refer to stand-ins for specific proxy records or particular proxy types. That is, the term pseudoproxy does not by itself imply that the modifications of the input data represent validly the uncertainties or characteristics of real world data. This view of the term pseudoproxy is in line with the past literature (compare, for example, Mann and Rutherford, 2002; Osborn and Briffa, 2004; von Storch et al., 2004; Jones et al., 2009; Graham and Wahl, 2011; Thompson et al., 2011; Lehner et al., 2012; Smerdon, 2012; Hind et al., 2012; Annan and Hargreaves, 2013; Kurahashi-Nakamura et al., 2014; Steiger and Hakim, 2016). These modifications may be simply by adding noise to the input data or may invoke more complex forward approaches (for example mechanistic Proxy System Models, Evans et al., 2013, see below).

-p2 l26-31: Considering dating uncertainty as purely additive white noise independent of the time axis strongly limits the suitability of the resulting time series. Autocorrelation results from the distortion of the time axis by changes in accumulation rate - which should, in a real proxy record, be captured by dating, and subsequent age modelling. Dating uncertainty represents a large component of the overall contribution to the low signal to noise ratio (c.f. Reschke, Rehfeld Laepple, Clim. Past. Discuss). The net cross-ensemble mean of the dating contribution to the final pseudoproxy uncertainty is zero in the presented formulation, as is the serial correlation of the component. Both is not appropriate. It would be beneficial to adopt (or include/prepare for) ensemblebased age models for the actual underlying proxy records; or if a simplistic solution is desired, to include the more realistic option of modeling age uncertainty by relative squeezing and stretching of the time axis.

We appreciate the referee's concerns. There are various points here. We disagree on the suitability of the resulting time-series. We are still confident that the time-series including the error terms are suitable for wide range of applications in, e.g., comparing different model simulations, model-data evaluation, and testing of reconstruction methods. The dating uncertainty in our set-up has two parts, the sampling of the dates and the sampling of the dating uncertainty on the one hand, and the associated dating error modelling on the other hand.

Indeed, not all our implementations of the dating error show notable serial correlation, however, most and particularly the main approach led to a small amount of serial correlation by making the dating uncertainty error dependent on previous errors and "measurements".

We tried to modify our code to allow for larger uncertainty and correlation in the final products. The changes do not result in notably increased serial correlations. We now provide data for this approach, but the initial approach is still included in the code. As mentioned above, we still regard the full set of pseudoproxies of immediate value for a number of tasks.

Thus, indeed, we choose to concentrate on a simplistic approach.

- p 3 l30 and following: Why is only summer seasonality considered? Is this a limitation of the pseudoproxy code?

In the initial submission, we concentrated on the modern boreal summer season since many authors attribute the sensitivity of their proxies to the summer season. We could have similarly used annual data or any other seasonal definition. The pseudoproxy code takes a time-series of any seasonality of interest. At this stage, it does not read the full annual matrix. We considered within the preparation of revisions to formulate the code more flexibly.

We now consider annual data for the simulation, and, in turn, also for the insolation bias. The code anyway allows to calculate the bias for arbitrary seasonal definitions, but we only consider one fixed definition for convenience. As noted, this is now annual.

- p4 l3: why this gridpoint? The arbitrariness of this choice somewhat illustrates that it appears difficult to use this code to include knowledge on real-life proxy datasets (e.g. sedimentation rate/dating frequency/ multi-proxy configurations).

As the referee already notes, this decision was generally arbitrary. It was made in relation to comparison with various proxies around the Iberian Peninsula and more generally Europe in another context. As we described, the visualisation for another proxy was provided in the Supplementary materials. As we state above, the aim here is not to test real world proxies but to provide data for testing methods and code to evaluate models and data. Our approach does not aim at mimicking real proxies but at providing pseudoproxies also for locations where we do not have real world knowledge.

We now consider a different grid point, which is located in the northwestern Pacific.

- Sec. 3: Please provide a graphical illustration of your pseudoproxy generation (e.g. using a graphical model).

We provide a visualisation of the procedure in the manuscript and in the manuscript assets.

- p6 l26: Autocorrelation should be considered, as several of the noise components (dating, non-local climate) are expected to be autocorrelated processes. The difficulty will be in actually estimating the true autocorrelation that should be used for the noise process.

As we note, the code provided allows for generating autocorrelated noise.

We now consider correlated noise at more instances.

- p7 l6: Is the assumption of increasing noise variability with increasing parameter variability appropriate for all noise components? It would appear that larger climatic variations might be recorded more precisely. In the absence of information whether proxy noise is smaller or larger for higher or lower climate variability this term should be reconsidered.

While larger variations may be recorded more precisely one may also assume that larger, e.g., temperature variations are associated with larger variations in other environmental components that may result in larger errors.

We introduced a switch.

- p7 l22: Can winter insolation be considered as bias?

If we understand the referees question correctly, then, yes, there is no reason why one should not consider winter insolation.

Our code does not calculate a winter insolation bias, but considering a winter insolation bias only requires changing a number of parameters defining the months of interest.

- p8 l5 and following: How do the processes and results here compare to the approach by Dolman and Laepple (2018)?

Our assumptions simplify the more complex approach of Dolman and Laepple. Our randomization could be seen as more realistic while the lack of process based assumptions make our approach less realistic.

Generally our results are not as smooth as the results of Dolman and Laepple considering the bioturbation while the assumptions on the subsampling appear to be comparable in our reading.

- p10 l13: typo original

We want to thank the referee for spotting this.

We change this in the revised version.

- p10 and following: The measurement error will depend on the type of sampling. To which degree is the sampling of the pseudo-proxy archive consecutive, overlapping, or spot-wise?

In this implementation, we only consider a conceptual measurement error and sample the time-series at certain years in the time-series as stated in the text.

We tried to clarify this in the text.

- p13 l12: Consider also the Bayesian Age-Depth modeling methods (e.g. OxCal, Bacon etc) which provide probability density functions of the proxy records.

We mention age modelling methods now, but do not discuss them in depth. We note, that we provide in principle most informations necessary to apply methods like Bchron (Haslett and Parnell, 2008) similar to their application in PRYSM by Dee et al. (2015, 2018). Additional information for these methods could be randomized.

- Figure 5: Please provide ensemble averages that allow to assess the spectral biases due to the proxy processes more easily.

We, at this point in the manuscript, do not use ensembles but only the single estimate.

We now use a wavelet based approach and weight the spectra to provide a smoother visualisation to ease the comparison.

- Figure 10: The time series are difficult to process and compare by eye. It appears in some cases there is an amplification of the apparent signal in the pseudoproxy record. Why? Where on the globe is the SD of the pseudoproxy > the SD of the climate signal?

This is mainly due to the size of the considered bias and the amplitude of noise processes.

While we announced to provide a visualisation equivalent to Figure 7 to answer the question about the SD, we decide simply to refer to the upper panels of Figure 7. These do not show the data from Figure 10 but highlight the SD-ratios

between input data and the proxies. We changed the visualisation there and now do not use the sampled interannual input data but samples of a 501-year moving average version of the input data. We also add a panel for the full sampled records.

- p28: missing section ref.

We want to thank the referee for spotting this.

We change this in the revised version.

[revised manuscript text omitted]

 height

**Table A3.** Continuation of list of parameters used.

[revised manuscript text omitted]

---

## Author Response (AR2)

Dear editor, dear referees,

Thank you for your time, effort, and the helpful suggestions.

Please find below our response.

On behalf of the authors

Sincerely yours,

Oliver Bothe

Referee 3:

First of all I would like to thank the authors for the clearly structured and useful reply document. The manuscript is much improved. I only have minor comments and one request for clarification regarding the insolation bias.

*Response:*
*Thank you for your efforts and these helpful corrections.*

- l. 3: sentence incorrect, correct by e.g. changing "and address their specific uncertainties" to "while addressing their specific uncertainties" or "and to address.."

*Response:*
*We change the sentence: "Similarly, comparing climate simulations and proxies requires approaches to bridge the temporal and spatial differences between both and to address their specific uncertainties."*

- l 19: ambiguously phrased: multiple variables influence the signal recorded (not multiple variables are recorded), please make more precise

*Response:*
*We change this to: "However, multiple variables influence the signal recorded, and we are often only interested or able to extract the contribution of one single climatic parameter."*

- l 21: "All other imprints of climate are noise on this variable of interest" -- "relative" implies a ratio

*Response:*
*We change the sentence to: "All other imprints of climate are noise with regard to this variable of interest."*

Page 10/11
- l. 15 and following: The insolation bias is computed from Insolation (given in W/m^2), and applied in ad hoc temperature units. Given that the annual mean is considered it is computed from some annual integral over the insolation at the proxy latitude. The bias is considered positive if the insolation is larger. (l. 25), however, the effective bias applied in Fig. 2 is positive at the LGM and negative in the early Holocene. Is this correct? Integrated insolation changes are the other way around (LGM is low, early Holocene high).

*Response:*
*The reviewer is correct, that the bias is a latitude specific annual value based on daily or monthly values. We are using a grid point at ~39°N. We use Crucifix (2016, R package 'palinsol') to compute the insolation data. The annual insolation is slightly larger at this latitude at the Last Glacial Maximum compared to modern insolation and both states have*

*larger insolation than the beginning of the Holocene. Marcott et al. (2013), Loutre et al. (2004), and Roche et al. (2011) provide visualisations of annual mean insolation by latitude against time. Below Figure R2_1 reproduces Figure 2(A) of Roche et al. (2011) without changes. The Figure is licensed under a CC-BY 3.0 license (https://creativecommons.org/licenses/by/3.0/).*

*We do not add or change anything in the manuscript in response to this comment.*

[Figure]

*Figure R2_1: Reproduction of Figure 2(A) of Roche et al. (2011). The Figure is licensed under a CC-BY 3.0 license (https://creativecommons.org/licenses/by/3.0/). The original caption reads for panel (A): Insolation anomaly to the 0–30 kyrs BP mean for the last deglaciation computed from Berger (1978) for (A) the annual mean.*

*Berger, A. L.: Long-term variations of caloric insolation resulting from earths orbital elements, Quaternary Res., 9, 139–167, 1978.*
*Crucifix, M. 2016, palinsol: Insolation for Palaeoclimate Studies, https://CRAN.R-project.org/package=palinsol*
*Loutre, M.-F., Paillard, D., Vimeux, F. and Cortijo, E.: Does mean annual insolation have the potential to change the climate?, Earth and Planetary Science Letters, 221(1–4), 1–14, doi:10.1016/s0012-821x(04)00108-6, 2004.*
*Marcott, S. A., Shakun, J. D., Clark, P. U. and Mix, A. C.: A Reconstruction of Regional and Global Temperature for the Past 11,300 Years, Science, 339(6124), 1198–1201, doi:10.1126/science.1228026, 2013.*
*Roche, D. M., Renssen, H., Paillard, D. and Levavasseur, G.: Deciphering the spatio-temporal complexity of climate change of the last deglaciation: a model analysis, Climate of the Past, 7(2), 591–602, doi:10.5194/cp-7-591-2011, 2011.*

- l2 missing ")"

*Response:*
*Thank you. We fix this.*

List of changes:

We address the technical corrections suggested by the referees. This results in edits to four sentences.

We further change the tex-\labels for Tables and Figures, which results in further tracked changes in the pdf.

[revised manuscript text omitted]